# GSK3-CRMP2 signaling mediates axonal regeneration induced by *Pten* knockout

Marco Leibinger[1], Alexander M. Hilla ᴵᴰ [1], Anastasia Andreadaki[1] & Dietmar Fischer ᴵᴰ [1]

Knockout of phosphatase and tensin homolog (PTEN$^{-/-}$) is neuroprotective and promotes axon regeneration in mature neurons. Elevation of mTOR activity in injured neurons has been proposed as the primary underlying mechanism. Here we demonstrate that PTEN$^{-/-}$ also abrogates the inhibitory activity of GSK3 on collapsin response mediator protein 2 (CRMP2) in retinal ganglion cell (RGC) axons. Moreover, maintenance of GSK3 activity in *Gsk3*$^{S/A}$ knockin mice significantly compromised PTEN$^{-/-}$-mediated optic nerve regeneration as well as the activity of CRMP2, and to a lesser extent, mTOR. These GSK3$^{S/A}$ mediated negative effects on regeneration were rescued by viral expression of constitutively active CRMP2$^{T/A}$, despite decreased mTOR activation. *Gsk3*$^{S/A}$ knockin or CRMP2 inhibition also decreased PTEN$^{-/-}$ mediated neurite growth of RGCs in culture and disinhibition towards CNS myelin. Thus, the GSK3/CRMP2 pathway is essential for PTEN$^{-/-}$ mediated axon regeneration. These new mechanistic insights may help to find novel strategies to promote axon regeneration.

[1] Department of Cell Physiology, Faculty of Biology and Biotechnology, Ruhr-University, 44780 Bochum, Germany. Correspondence and requests for materials should be addressed to D.F. (email: dietmar.fischer@rub.de)

Mature neurons of the central nervous system (CNS) cannot normally regenerate injured axons due to an inhibitory environment of axonal growth cones and an insufficient intrinsic growth ability[1–3]. Therefore, CNS injuries are often associated with an irreversible functional loss, such as blindness after optic nerve injury or paralysis and sensation loss after spinal cord damage. However, research over the last two decades has developed several approaches to improve the intrinsic ability of adult CNS neurons to regrow axons. For instance, genetic or shRNA-mediated neuronal depletion of phosphatase and tensin homolog (PTEN), which activates the phosphoinositide 3-kinase/protein kinase B (PI3K/AKT) pathway in respective neurons, enables axon regeneration over considerable distances into the injured optic nerve or spinal cord[4–8]. In the optic nerve, these effects can be mostly mimicked by expression of constitutively active AKT[9,10]. Active AKT modulates several downstream regulators by phosphorylation, such as tuberous sclerosis complex (TSC) or glycogen synthase kinase 3 (GSK3)[9,11–14]. Regarding *Pten* knockout (PTEN$^{-/-}$)-mediated axon regeneration, the PI3K/AKT/TSC signaling pathway, which activates the mammalian target of rapamycin (mTOR) has been made accountable for the strong neuroprotective as well as axon regeneration promoting effects. Consistently, treatment with the (mTOR) inhibitor rapamycin partially compromises the beneficial effect induced by PTEN$^{-/-}$ on neuroprotection as well as regeneration[4]. However, targeting effectors directly downstream or even regulators upstream of mTOR, such as S6 kinase1 (S6K1), E4 binding protein (E4BP) or Ras homolog enriched in brain (Rheb), were able to induce neuroprotection, but failed to mimic the entire PTEN$^{-/-}$ effect on axon regeneration[15–18]. Interestingly, a recent study even proposed an adverse effect of S6K1 activity for spinal cord axon regeneration[19]. These reports suggest that the regenerative effects of PTEN$^{-/-}$ do not solely depend on mTOR, but also other downstream effectors of AKT.

Another approach to promote optic nerve regeneration is the stimulation of RGCs by cytokines of the interleukin 6 (IL-6) superfamily, which are released from activated retinal glial cells after an inflammatory stimulation (IS) induced by lens injury[20,21]. IS activates the JAK/STAT3 pathway but has little effect on the PI3K/AKT pathway[22–26]. Genetic or shRNA-mediated conditional knockdown of GSK3β (GSK3β$^{-/-}$) markedly promotes IS-induced optic nerve regeneration by reducing inhibitory phosphorylation of axonal collapsin response mediator protein 2 (CRMP2) in RGCs, demonstrating that GSK3/CRMP2 signaling is highly relevant for optic nerve regeneration and most likely also other parts of the CNS[27].

As GSK3 is a substrate of AKT, the current study tested the hypothesis that the GSK3/CRMP2 pathway could be an essential part of PTEN$^{-/-}$-mediated axon regeneration. We found that PTEN$^{-/-}$ releases CRMP2 from GSK3-dependent inhibition in RGC axons. Moreover, PTEN$^{-/-}$ mediated optic nerve regeneration was significantly reduced by inhibitory CRMP2 phosphorylation in *Gsk3*$^{S/A}$ knockin mice, but rescued by viral expression of constitutively active CRMP2$^{T/A}$, indicating its essential role in this context. As the number of publications using PTEN depletion mediated neuronal regeneration is still increasing[28], these mechanistic findings are highly relevant for data interpretation and may facilitate new regenerative strategies mitigating the high cancerogenic risk of PTEN$^{-/-}$ or inhibition in the future.

## Results

### Neuronal PTEN$^{-/-}$ induces inhibitory GSK3 phosphorylation.

Optic nerve crush (ONC) causes moderate AKT activation and subsequent inhibitory phosphorylation of GSK3α (phospho-serine-21-GSK3α, pGSK3α) and GSK3β (phospho-serine-9-GSK3β, pGSK3β) in RGCs of adult mice[27]. As neuronal *Pten* knockout (PTEN$^{-/-}$) triggers AKT activation much stronger than ONC (Fig. 1a–d), we first investigated whether it would also induce inhibitory phosphorylation of both GSK3 isoforms. To this end, *Pten* floxed mice received intravitreal injections of either AAV-Cre to specifically delete *Pten* in RGCs (PTEN$^{-/-}$) or respective control virus (AAV-GFP) (PTEN$^{+/+}$). Some animals were subjected to additional ONC 3 weeks after viral application. Consistent with previous data[27], retinae of uninjured PTEN$^{+/+}$ mice showed only a few pGSK3α- and pGSK3β-positive RGCs in retinal cross-sections, while ONC significantly increased numbers and intensities of positively stained RGCs when evaluated 5 days after surgery (Fig. 1b, c). PTEN$^{-/-}$ alone induced much stronger pGSK3α- and pGSK3β-levels in RGCs compared with axotomy in PTEN$^{+/+}$ mice, while PTEN$^{-/-}$ +ONC showed no additive effect (Fig. 1b, c). The absence of pGSK3α and pGSK3β signals in retinal cross sections of respective non-phosphorylatable *Gsk3α*$^{S/A}$ or *Gsk3β*$^{S/A}$ knockin mice[27,29] verified the staining specificity of both pGSK3 antibodies (Fig. 1b, c). Western blot analyses of retinal lysates from similarly treated mice confirmed the upregulation of GSK3 phosphorylation upon PTEN$^{-/-}$, while total levels of both isoforms remained unchanged (Fig. 1d–h, Supplementary Fig. 1a, d). AKT phosphorylation was induced markedly stronger by PTEN$^{-/-}$ compared with ONC alone (Fig. 1d, i) and neither AKT nor GSK3 phosphorylation was further increased after additional ONC (Fig. 1d–f, i, Supplementary Fig. 1b, c, f), thereby verifying our immunohistochemical data. Moreover, retinal levels of phosphorylated ribosomal protein S6 (pS6), an indicator for mTOR activity, were markedly elevated after PTEN$^{-/-}$ and were not reduced by ONC, whereas they were in PTEN$^{+/+}$ controls (Fig. 1d, j, Supplementary Fig. 1e). Thus, besides increasing and maintaining mTOR activity, PTEN$^{-/-}$ lead to significant inhibitory phosphorylation of GSK3 in RGCs.

To test the possibility of whether ONC- or *Pten* knockout-stimulated GSK3 inhibition was mTOR activity-dependent (Fig. 2a), we treated wild type or PTEN$^{-/-}$ mice with either vehicle or the mTOR inhibitor rapamycin in vivo. Western blot analysis and immunohistochemistry showed that despite the expected absence of pS6 in rapamycin-treated mice, total GSK3 expression as well as ONC- or PTEN$^{-/-}$-induced pGSK3α and pGSK3β levels were similar across retinae of all uninjured and ONC-treated wild type and PTEN$^{-/-}$ animals (Fig. 2b–g, Supplementary Fig. 2a–e, Supplementary Fig. 3a–d). Therefore, GSK3 phosphorylation in RGCs is mTOR activity independent.

### Active GSK3 compromises PTEN$^{-/-}$-induced axon regeneration.

We next tested whether PTEN$^{-/-}$-induced GSK3 inhibition could be part of the axon regeneration promoting mechanism. To this end, *Pten*$^{f/f}$ animals were crossbred with mice carrying different constitutively active forms of GSK3 (GSK3α$^{S/A}$, GSK3β$^{S/A}$) or wild-types and received intravitreal injections of either AAV-GFP (PTEN$^{+/+}$) or AAV-Cre (PTEN$^{-/-}$). Three weeks after that, all animals were subjected to ONC. The resulting regenerative states of RGCs were evaluated either by quantification of spontaneous neurite growth in cultures that were prepared 5 days after ONC (Fig. 3a, b)[27,30] or by quantification of regenerating axons beyond the lesion site of the nerve 3 weeks after ONC (Fig. 3c–h). Expectedly, PTEN$^{-/-}$ markedly increased spontaneous neurite growth of cultured RGCs (Fig. 3a, b) and regeneration of anterogradely labeled axons in the optic nerve in vivo compared with PTEN$^{+/+}$ animals (Fig. 3c–h, Supplementary Fig. 4a, b). Notably, the PTEN$^{-/-}$ effects on neurite growth in culture, as well as

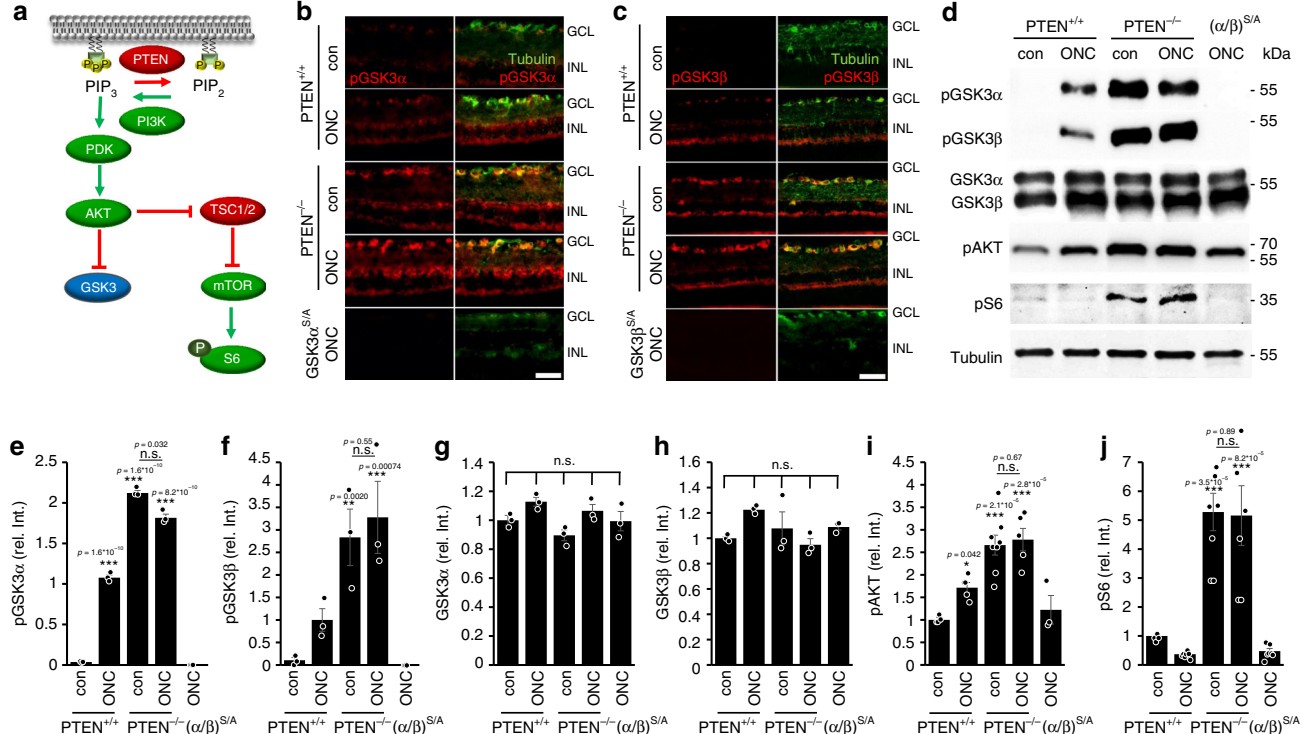

**Fig. 1** PTEN$^{-/-}$ induces inhibitory GSK3 phosphorylation. **a** Schematic drawing illustrating the PTEN/AKT/mTOR/pS6 and PTEN/AKT/GSK3 signaling pathways. PTEN depletion activates phosphatidylinositol 3-kinase (PI3K), which then converts phosphatidylinositol (4, 5) bisphosphate (PIP$_2$) into phosphatidylinositol (3, 4, 5) trisphosphate (PIP$_3$). The latter activates AKT by a phosphatidylinositol-dependent kinase (PDK) 1/2-mediated phosphorylation. AKT activates mammalian target of rapamycin (mTOR) by releasing TSC1/2 inhibition leading to phosphorylation of ribosomal protein S6 via mTOR. Moreover, it inhibits GSK3 isoforms by phosphorylation. **b**, **c** Retinal cross-sections of PTEN$^{f/f}$ mice after intravitreal AAV-GFP (PTEN$^{+/+}$) or AAV-Cre injection leading to PTEN depletion (PTEN$^{-/-}$). Animals remained either untreated (con) or received ONC 5 days before tissue isolation. ONC-induced S21-phosphorylation of GSK3α (pGSK3α, red, **b**) and S9-phosphorylation of GSK3β (pGSK3β, red, **c**) in βIII-tubulin-positive RGCs (green) in PTEN$^{+/+}$, while sole PTEN$^{-/-}$ showed already a markedly stronger effect. Inhibitory GSK3 phosphorylation was absent in retinae of Gsk3α$^{S/A}$, or Gsk3β$^{S/A}$ knockin mice, verifying antibody specificity. GCL, ganglion cell layer; INL, inner nuclear layer. Scale bars: 50 μm. **d** Western blots: Retinal lysates from animals as described in (**b**) and (**c**). ONC moderately increased pGSK3α and pGSK3β signals in PTEN$^{+/+}$ mice compared with untreated controls, whereas PTEN$^{-/-}$ showed a much stronger effect. Lysates from Gsk3(α/β)$^{S/A}$ knockin ((α/β)$^{S/A}$) mice showed no pGSK3 signals, verifying antibody specificities. Levels of T$^{308}$-phosphorylated AKT (pAKT) were elevated after ONC in PTEN$^{+/+}$ and (α/β)$^{S/A}$ mice, but stronger increased by PTEN$^{-/-}$. Phosphorylation of ribosomal protein S6 (pS6) was low in the PTEN$^{+/+}$ retinae and further decreased by ONC, while PTEN$^{-/-}$ induced a strong upregulation in uninjured and injured retinae. Total GSK3α and GSK3β levels were comparable in all experimental groups. Loading control: βIII-tubulin (tubulin). **e-j** Quantification of Western blots in (**d**) relative to βIII-tubulin and normalized to PTEN$^{+/+}$ con, or in case of pGSK3α and pGSK3β to ONC. One-way analysis of variance (ANOVA) with Tukey post hoc test was used. Treatment effects compared with PTEN$^{+/+}$ con: **$p < 0.01$; ***$p < 0.001$; n.s. = non-significant. Values represent means ± SEM of 3–8 retinae per group (**e-h**: $n = 3$; **i**: PTEN$^{+/+}$ con, $n = 4$; PTEN$^{+/+}$ ONC, $n = 5$; PTEN$^{-/-}$ con, $n = 8$; PTEN$^{-/-}$ ONC, $n = 5$; PTEN$^{-/-}$(α/β$^{S/A}$) ONC, $n = 3$; **j**: PTEN$^{+/+}$ con, $n = 4$; PTEN$^{+/+}$ ONC, $n = 7$; PTEN$^{-/-}$ con, $n = 7$; PTEN$^{-/-}$ ONC, $n = 6$; PTEN$^{-/-}$(α/β$^{S/A}$) ONC, $n = 7$).

regeneration in vivo were significantly reduced in RGCs from both Pten$^{-/-}$/Gsk3$^{S/A}$ knockin mice (PTEN$^{-/-}$/GSK3α$^{S/A}$, PTEN$^{-/-}$/GSK3β$^{S/A}$) (Fig. 3a–h, Supplementary Fig. 4c, d). PTEN$^{-/-}$-induced neuroprotection was not affected by Gsk3α$^{S/A}$ or Gsk3β$^{S/A}$ knockin when evaluated 3 weeks after ONC (Fig. 4a, b), demonstrating that these effects were solely restricted to axon growth and regeneration.

**Inhibition of GSK3 activates mTOR in neurons.** In non-neuronal cells, active GSK3 reportedly compromises mTOR activity via activating phosphorylation of tuberous sclerosis complex 2 (TSC2)[31]. We, therefore, explored the possibility of whether GSK3β inhibition (phosphorylation) may indirectly contribute to the effect of PTEN$^{-/-}$ on mTOR activity (Fig. 5a). To this end, we immunohistochemically stained retinal flat mounts of untreated or axotomized (ONC) wt, GSK3β$^{S/A}$, conditional PTEN$^{-/-}$, or PTEN$^{-/-}$/GSK3β$^{S/A}$ mice for phospho-S6

(pS6) to determine the mTOR activity in RGCs[4,15]. Consistent with previous reports[15,32], co-staining against βIII-tubulin showed ~15% of pS6-positive RGCs in wt retinae and an increase in the number and intensity of pS6-positive RGCs (~40%) as well as increased RGC soma size in PTEN$^{-/-}$ mice which was previously described as a consequence of S6K1 activity downstream of mTOR[15] (Fig. 5b, c). While ONC significantly decreased the percentage of pS6-positive RGCs in wt mice, it did not affect the high numbers in animals with PTEN$^{-/-}$ (Fig. 5b, c). Interestingly, GSK3β$^{S/A}$ compromised PTEN$^{-/-}$-mediated mTOR activation in retinal flat mounts of both, uninjured and injured mice indicated by reduced numbers of pS6-positive RGCs (Fig. 5b, c). Consistently, Western blot analysis showed reduced pS6 levels in retinal lysates from PTEN$^{-/-}$/GSK3β$^{S/A}$ mice compared with lysates from just PTEN$^{-/-}$ animals despite similarly enhanced pAKT in both genotypes (Fig. 5d–f, Supplementary Fig. 5a–c). However, Gsk3β$^{S/A}$ knockin

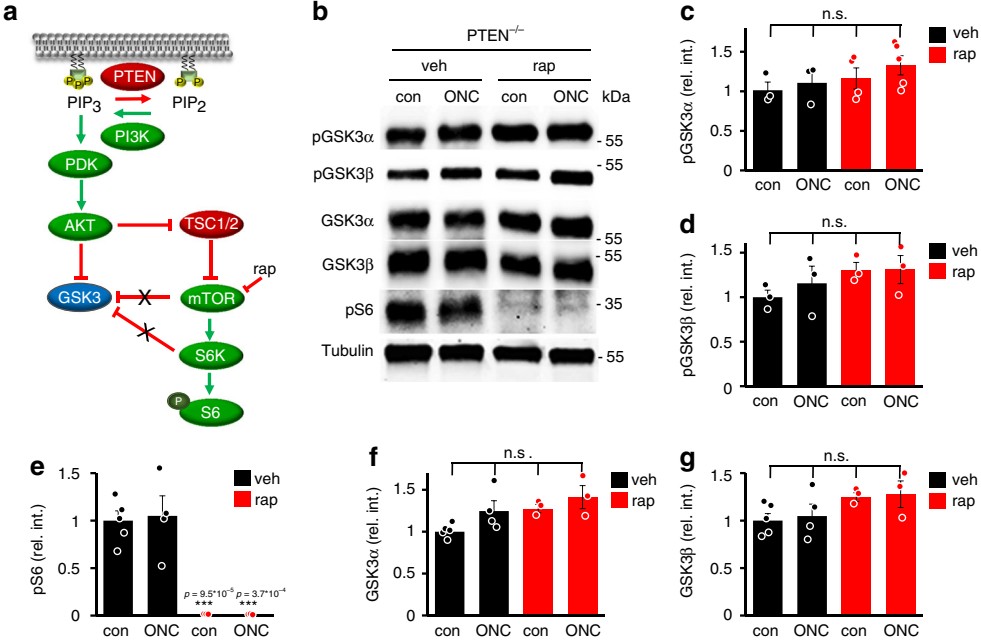

**Fig. 2** PTEN$^{-/-}$-mediated GSK3 inhibition is mTOR independent. **a** Schematic drawing illustrating the outcome of experiments shown in (**b-g**). **b** Western blots: Retinal lysates from PTEN$^{-/-}$ mice either uninjured (con) or 5 days after optic nerve crush (ONC) and intraperitoneal injection with either vehicle (veh) or rapamycin (rap). PTEN$^{-/-}$-induced phospho-GSK3α (pGSK3α) and phospho-GSK3β (pGSK3β) levels were not affected by rapamycin treatment. Absent pS6 levels in rapamycin-treated samples verified efficient mTOR inhibition. βIII-tubulin (tubulin) served as a loading control. **c-g** Densitometric quantifications of Western blots as depicted in (**b**) relative to βIII-tubulin and normalized to vehicle con. Significances of intergroup differences were evaluated using two-way analysis of variance (ANOVA) with Tukey post hoc test. Treatment effects compared with wt con: ***$p < 0.001$; n.s. = non-significant. Values represent means ± SEM of 3–5 retinae per group (**c**: veh con, $n = 3$; veh ONC, $n = 3$; rap con, $n = 4$; rap ONC, $n = 6$; **d**: $n = 3$; **e**: veh con, $n = 5$; veh ONC, $n = 4$; rap con, $n = 3$; rap ONC, $n = 3$; **f**: veh con, $n = 5$; veh ONC, $n = 4$; rap con, $n = 3$; rap ONC, $n = 3$; **g**: veh con, $n = 5$; veh ONC, $n = 4$; rap con, $n = 3$; rap ONC, $n = 3$)

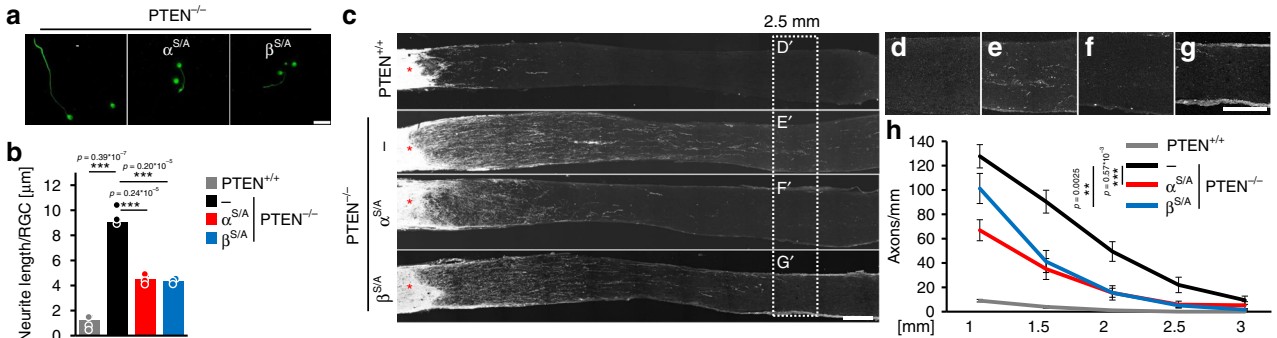

**Fig. 3** GSK3$^{S/A}$ compromises PTEN-induced optic nerve regeneration. **a** βIII-tubulin-positive RGCs from PTEN$^{-/-}$ mice with either *Gsk3α$^{S/A}$* (α$^{S/A}$), *Gsk3β$^{S/A}$* (β$^{S/A}$), or no (−) knockin after 48 h in culture. Animals received ONC 5 days before culture preparation to transform neurons into a regenerative state. Scale bar: 50 μm. **b** Quantification of spontaneous neurite growth in cultures as described in (**a**). Knockins of α$^{S/A}$ and β$^{S/A}$ markedly compromised PTEN$^{-/-}$ triggered neurite growth of RGCs. Values represent means ± SEM from three independent experiments ($n = 3$) with four wells as technical replicates per group. Neurite length per RGC refers to a total number of ~500 RGCs per well with ~30 RGCs showing neurite growth. **c** Representative longitudinal optic nerve sections from mice with genotypes as described in (**a**) with regenerating, CTB-labeled axons 21 days after ONC. Lesion sites are indicated by asterisks. Scale bar: 200 μm. **d-g** Higher magnifications of respective optic nerve sections 2.5 mm distal to the injury site as indicated in (**c**). Scale bar: 200 μm. **h** Quantification of regenerating axons per mm optic nerve width at 1, 1.5, 2, 2.5, and 3 mm beyond the lesion site in treatment groups as in (**c**). PTEN$^{-/-}$-triggered optic nerve regeneration was significantly compromised in α$^{S/A}$ and β$^{S/A}$ knockin compared with wt mice. Values represent means ± SEM of 4–8 animals (PTEN$^{+/+}$ $n = 4$; PTEN$^{-/-}$ $n = 8$; PTEN$^{-/-}$/α$^{S/A}$ $n = 7$, PTEN$^{-/-}$/β$^{S/A}$ $n = 8$) per experimental group. Significances of intergroup differences in (**b**) and (**h**) were evaluated using one-way analysis of variance (ANOVA) with Holm-Sidak post hoc test. Treatment effects: **$p < 0.01$, ***$p < 0.001$

was not sufficient to reduce RGC size in PTEN$^{-/-}$ mice, while *Gsk3(α/β)* double knockin did so (Supplementray Fig. 6a, b).

We then tested whether *Gsk3α* and/or *Gsk3β* knockouts have the opposite effect. To this end, we generated *Gsk3β$^{f/f}$/Pten$^{f/f}$*

mice, which along with *Gsk3α$^{f/f}$, Gsk3β$^{f/f}$, Gsk3(α/β)$^{f/f}$*, and *Pten$^{f/f}$* animals received intravitreal injections of either AAV-Cre or AAV-GFP resulting in either PTEN$^{-/-}$, GSK3α$^{-/-}$, GSK3β$^{-/-}$, GSK3(α/β)$^{-/-}$, GSK3β$^{-/-}$/PTEN$^{-/-}$, or GSK3β$^{+/+}$/PTEN$^{+/+}$

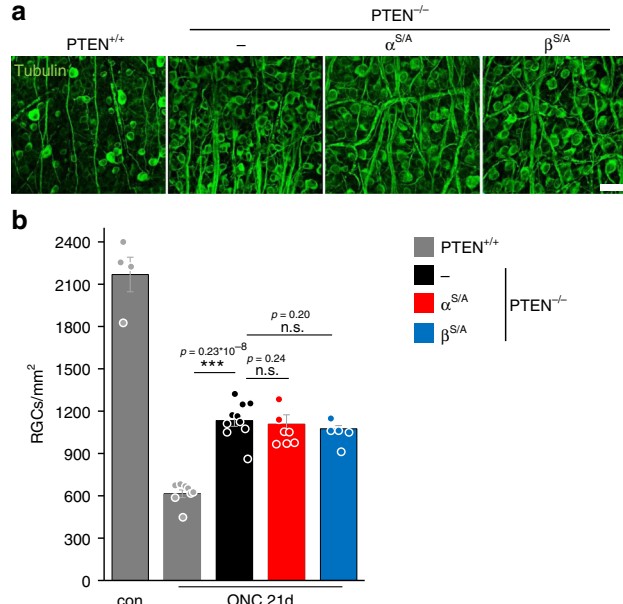

**Fig. 4** GSK3$^{S/A}$ does not affect PTEN$^{-/-}$-induced neuroprotection. **a** βIII-tubulin-stained whole-mount retinae from PTEN$^{+/+}$ mice and animals with genotypes, as described in Fig. 3a, 21 days after ONC. Scale bar: 50 μm. **b** Quantification of RGCs per mm$^2$ in retinal whole-mounts as described in (**a**) and untreated PTEN$^{+/+}$ mice (con). The numbers of surviving RGCs 21 days after ONC were similar for the respective PTEN$^{-/-}$ and PTEN$^{-/-}$/GSK3$^{S/A}$ genotypes. Significances of intergroup differences were evaluated using one-way analysis of variance (ANOVA) with Holm-Sidak post hoc test. Treatment effects: ***$p < 0.001$; n.s. = non-significant. Values represent means ± SEM of 4–10 retinae per group (con, $n = 4$; PTEN$^{+/+}$, $n = 8$; PTEN$^{-/-}$, $n = 10$; PTEN$^{-/-}$/α$^{S/A}$, $n = 7$; PTEN$^{-/-}$/β$^{S/A}$, $n = 5$)

(wt) RGCs. Half of the animals were additionally subjected to ONC and their retinae collected after 5 days. Consistent with previous reports[4,32], ONC reduced mTOR activity in wt animals (Fig. 6a, b). While GSK3α$^{-/-}$ did not upregulate mTOR activity (Fig. 6a, b), GSK3β$^{-/-}$ alone slightly increased pS6 levels in RGCs. Moreover, GSK3α$^{-/-}$, GSK3β$^{-/-}$, and PTEN$^{-/-}$ prevented an axotomy-induced downregulation of mTOR activity (Fig. 6a, b). GSK3(α/β)$^{-/-}$ showed even stronger S6 phosphorylation than every single knockout alone, but did not reach the full effects of PTEN$^{-/-}$ (Fig. 6a, b). In addition, only GSK3(α/β)$^{-/-}$ but not the knockout of single isoforms led to an increase in RGC soma size (Supplementary Fig. 6c, d). Phospho-S6 levels were not further enhanced by PTEN$^{-/-}$/GSK3β$^{-/-}$ (Fig. 6a, b), suggesting that PTEN$^{-/-}$ alone already maximally stimulates mTOR. Regarding RGC survival, GSK3α$^{-/-}$ and GSK3β$^{-/-}$ alone were not neuroprotective (Fig. 6c, d). However, the number of surviving RGCs was significantly increased by GSK3(α/β)$^{-/-}$, which was abolished by the mTOR inhibitor rapamycin (Fig. 6c, d), indicating a correlation between mTOR activity and neuroprotection.

To test whether GSK3 might affect S6 phosphorylation downstream from mTOR (Fig. 7a) we treated *Gsk3β*$^{-/-}$ mice with either vehicle or rapamycin. In addition, half of the animals were subjected to ONC 5 days before tissue isolation. Rapamycin treatment completely abrogated GSK3β$^{-/-}$-induced S6 phosphorylation (Fig. 7b, c) as it did after PTEN$^{-/-}$ in a similar experimental setting (Fig. 7d, e), indicating that GSK3β acts upstream of mTOR. Taken together these data lend support to the notion that PTEN$^{-/-}$-induced mTOR activation partially depends on its inhibitory effect on GSK3.

**PTEN$^{-/-}$-mediated CRMP2 activation is GSK3 dependent.** Inhibitory phosphorylation of CRMP2 at threonine 514 (T$^{514}$) by GSK3β (Fig. 8a) compromises, while prevention of

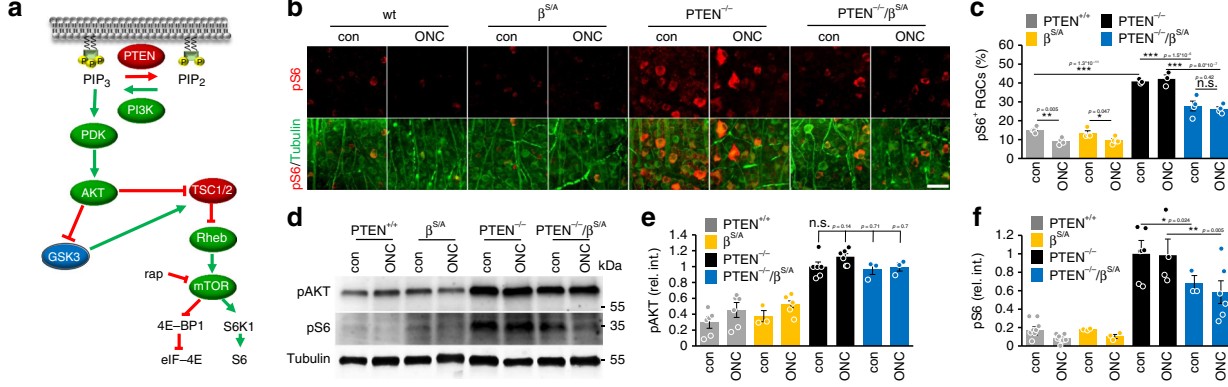

**Fig. 5** PTEN$^{-/-}$-induced mTOR activity depends on GSK3 inhibition. **a** Schematic drawing illustrating the outcome of experiments depicted in Figs. 5 and 6. **b** Retinal flat mounts isolated from PTEN$^{+/+}$/GSK3(α/β)$^{+/+}$ (wt), GSK3β$^{S/A}$ (β$^{S/A}$), PTEN$^{-/-}$, and PTEN$^{-/-}$/GSK3β$^{S/A}$ (PTEN$^{-/-}$/ β$^{S/A}$) mice which were either left untreated (con) or subjected to optic nerve crush (ONC). Five days after surgery, retinae were fixed and stained for phosphorylated ribosomal protein S6 (pS6; red) and βIII-tubulin (tubulin; green in lower row to visualize RGCs). Scale bar: 50 μm. **c** Quantification of pS6-positive RGCs in flat mounts as depicted in (**b**). Compared with wt animals, pS6 levels were markedly increased after PTEN$^{-/-}$ and significantly compromised in animals with additional GSK3β$^{S/A}$. Values represent means ± SEM of 3–5 retinae per group (untreated controls: PTEN$^{+/+}$, $n = 4$; β$^{S/A}$, $n = 4$; PTEN$^{-/-}$, $n = 3$; PTEN$^{-/-}$/β$^{S/A}$, $n = 4$; mice 5 days after ONC: PTEN$^{+/+}$, $n = 4$; β$^{S/A}$, $n = 5$; PTEN$^{-/-}$, $n = 3$; PTEN$^{-/-}$/β$^{S/A}$, $n = 4$). **d** Western blot analysis for phosphorylated AKT (pAKT) and pS6 of retinal lysates from animals as described in (**b**). βIII-tubulin (tubulin) served as a loading control. **e, f** Densitometric quantification of Western blots as depicted in (**d**) relative to βIII-tubulin and normalized to PTEN$^{-/-}$ con. Values represent means ± SEM of 3–8 retinae per group (**e**: untreated controls: PTEN$^{+/+}$, $n = 6$; β$^{S/A}$, $n = 3$; PTEN$^{-/-}$, $n = 6$; PTEN$^{-/-}$/β$^{S/A}$, $n = 3$; mice 5 days after ONC: PTEN$^{+/+}$, $n = 6$; β$^{S/A}$, $n = 5$, PTEN$^{-/-}$, $n = 6$; PTEN$^{-/-}$/β$^{S/A}$, $n = 3$; **f**: untreated controls: PTEN$^{+/+}$, $n = 8$; β$^{S/A}$ $n = 3$; PTEN$^{-/-}$, $n = 7$; PTEN$^{-/-}$/β$^{S/A}$, $n = 3$; mice 5 days after ONC: PTEN$^{+/+}$, $n = 7$; β$^{S/A}$ $n = 3$; PTEN$^{-/-}$, $n = 4$; PTEN$^{-/-}$/β$^{S/A}$, $n = 6$). Significances of intergroup differences were evaluated using two-way analysis of variance (ANOVA) with Tukey or Holm-Sidak post hoc test. Treatment effects to PTEN$^{+/+}$ con, or as indicated: *$p < 0.05$, **$p < 0.01$, ***$p < 0.001$, n.s. = non-significant

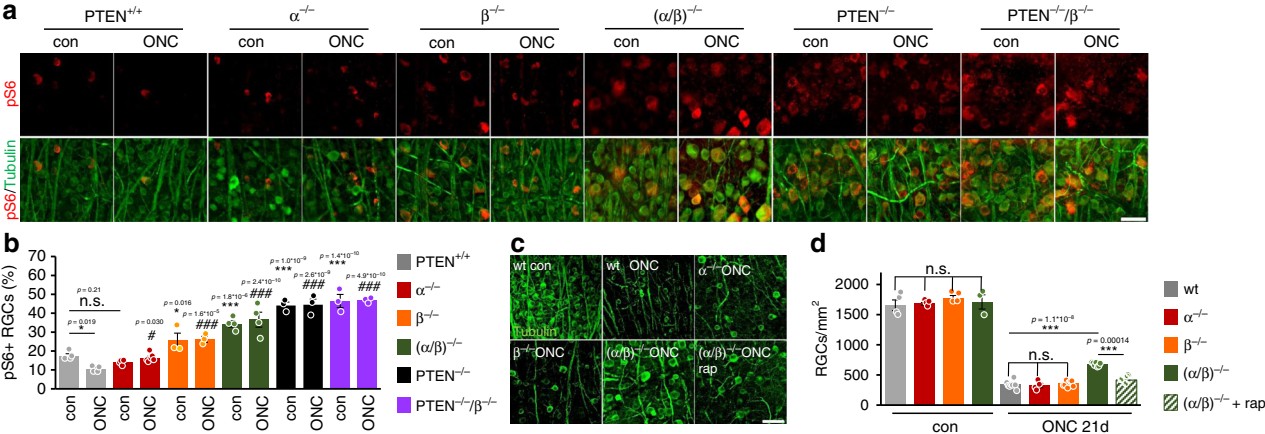

**Fig. 6** PTEN$^{-/-}$-induced mTOR activity depends on GSK3 inhibition. **a** Retinal flat mounts isolated from either PTEN$^{+/+}$/GSK3($\alpha$/$\beta$)$^{+/+}$ (PTEN$^{+/+}$), PTEN$^{+/+}$/GSK3$\alpha^{-/-}$ ($\alpha^{-/-}$), PTEN$^{+/+}$/GSK3$\beta^{-/-}$ ($\beta^{-/-}$), PTEN$^{+/+}$/GSK3($\alpha$/$\beta$)$^{-/-}$ (($\alpha$/$\beta$)$^{-/-}$), PTEN$^{-/-}$/GSK3($\alpha$/$\beta$)$^{+/+}$ (PTEN$^{-/-}$), or PTEN$^{-/-}$/ GSK3$\beta^{-/-}$ (PTEN$^{-/-}$/$\beta^{-/-}$) mice that were either left untreated (con) or subjected to ONC. Five days after surgery, retinae were stained as indicated. Scale bar: 50 μm. **b** Quantification of the percentage of pS6-positive RGCs as depicted in (**a**). Values represent means ± SEM of 2–6 retinae per group (untreated controls: PTEN$^{+/+}$, $n = 4$; $\alpha^{-/-}$ $n = 4$; $\beta^{-/-}$, $n = 3$; ($\alpha$/$\beta$)$^{-/-}$ $n = 4$; PTEN$^{-/-}$, $n = 3$; PTEN$^{-/-}$/$\beta^{-/-}$, $n = 3$; mice 5 days after ONC: PTEN$^{+/+}$, $n = 4$; $\alpha^{-/-}$, $n = 6$; $\beta^{-/-}$, $n = 3$; ($\alpha$/$\beta$)$^{-/-}$, $n = 4$; PTEN$^{-/-}$, $n = 3$; PTEN$^{-/-}$/$\beta^{-/-}$, $n = 3$). **c** βIII-tubulin-stained whole-mount retina from GSK3($\alpha$/$\beta$)$^{+/+}$ (wt), $\alpha^{-/-}$, $\beta^{-/-}$, or ($\alpha$/$\beta$)$^{-/-}$ animals systemically treated with the mTOR inhibitor rapamycin (rap). Mice were either untreated (con), or subjected to ONC and sacrificed 21 days after that. Scale bar: 50 μm. **d** Quantification of RGCs per mm$^2$ in retinal whole-mounts from mice as described in (**c**) and untreated $\alpha^{-/-}$, $\beta^{-/-}$, ($\alpha$/$\beta$)$^{-/-}$ mice. The number of surviving RGCs were significantly increased in ($\alpha$/$\beta$)$^{-/-}$ mice, which was almost entirely abolished by rapamycin treatment. Values represent means ± SEM of 3–8 retinae per group (untreated controls: wt, $n = 4$; $\alpha^{-/-}$, $n = 4$; $\beta^{-/-}$, $n = 4$; ($\alpha$/$\beta$)$^{-/-}$, $n = 3$; mice 5 days after ONC: wt, $n = 8$; $\alpha^{-/-}$, $n = 4$; $\beta^{-/-}$, $n = 8$; ($\alpha$/$\beta$)$^{-/-}$, $n = 7$; ($\alpha$/$\beta$)$^{-/-}$ +rap, $n = 3$). Significances of intergroup differences were evaluated using two-way (**b**), or one-way (**d**) analysis of variance (ANOVA) with Tukey or Holm-Sidak post hoc test. Treatment effects to PTEN$^{+/+}$ con, or as indicated: *$p < 0.05$, **$p < 0.01$, ***$p < 0.001$, n.s. = non-significant. Treatment effects to PTEN$^{+/+}$ ONC: #$p < 0.05$, ###$p < 0.001$

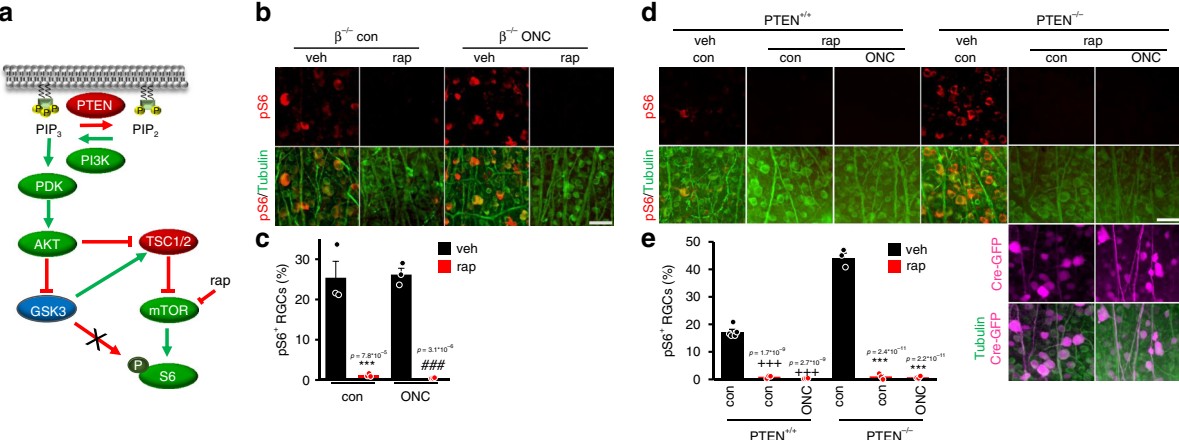

**Fig. 7** GSK3$\beta^{-/-}$-induced S6 phosphorylation is mTOR-dependent. **a** Schematic illustration of results shown in experiments of **b**–**e**. **b** Retinal flat mounts isolated from GSK3$\beta^{-/-}$ mice that were either left untreated (con) or subjected to ONC and, additionally, received a systemic injection of either vehicle (veh) or rapamycin (rap). Five days after injury, retinae were stained for phosphorylated S6 (pS6; red) and βIII-tubulin (tubulin; green in a lower row to visualize RGCs). Scale bar: 50 μm. **c** Quantification of pS6-positive RGCs as depicted in **b**. Rapamycin abolished $\beta^{-/-}$-induced S6 phosphorylation. Values represent means ± SEM of three retinae per group ($n = 3$). **d** Retinae isolated from PTEN$^{+/+}$ and PTEN$^{-/-}$ mice that were either left untreated (con) or subjected to ONC and, additionally, received a systemic injection of either veh or rap. Five days after surgery, retinae were stained for pS6 and tubulin. GFP (Cre-gfp, magenta) staining was applied to visualize transduction with GFP co-expressing AAV-Cre. Scale bar: 50 μm. **e** Quantification of pS6-positive RGCs as depicted in (**d**). Rapamycin abolished PTEN$^{-/-}$-induced S6 phosphorylation. Values represent means ± SEM of 3–5 retinae per group (PTEN$^{+/+}$ con, $n = 5$; PTEN$^{+/+}$ con + rap, $n = 3$; PTEN$^{+/+}$ ONC + rap, $n = 3$; PTEN$^{-/-}$ con, $n = 3$; PTEN$^{-/-}$ con + rap, $n = 3$; PTEN$^{-/-}$ ONC + rap, $n = 3$). Significances of intergroup differences in (**c**) and (**e**) were evaluated using two-way analysis of variance (ANOVA) with Holm-Sidak post hoc test. Treatment effects compared with $\beta^{-/-}$ veh con: ***$p < 0.001$, compared with $\beta^{-/-}$ veh ONC: ###$p < 0.001$, compared with PTEN$^{+/+}$ veh con: +++$p < 0.001$, or compared with PTEN$^{-/-}$ veh con: ***$p < 0.001$

phosphorylation promotes axon growth and overcomes myelin inhibition[27,33]. To examine a potential contribution of inhibitory CRMP2 phosphorylation in PTEN$^{-/-}$ stimulated regeneration we analyzed phospho-T$^{514}$-CRMP2 (pCRMP2) levels in proximal

segments of optic nerves (either crushed or uninjured) from wt, PTEN$^{-/-}$, and the different PTEN$^{-/-}$/GSK3$^{S/A}$ genotypes (PTEN$^{-/-}$/GSK3$\alpha^{S/A}$, or PTEN$^{-/-}$/GSK3$\beta^{S/A}$) as described previously[27]. Western blot analyses of nerve lysates showed

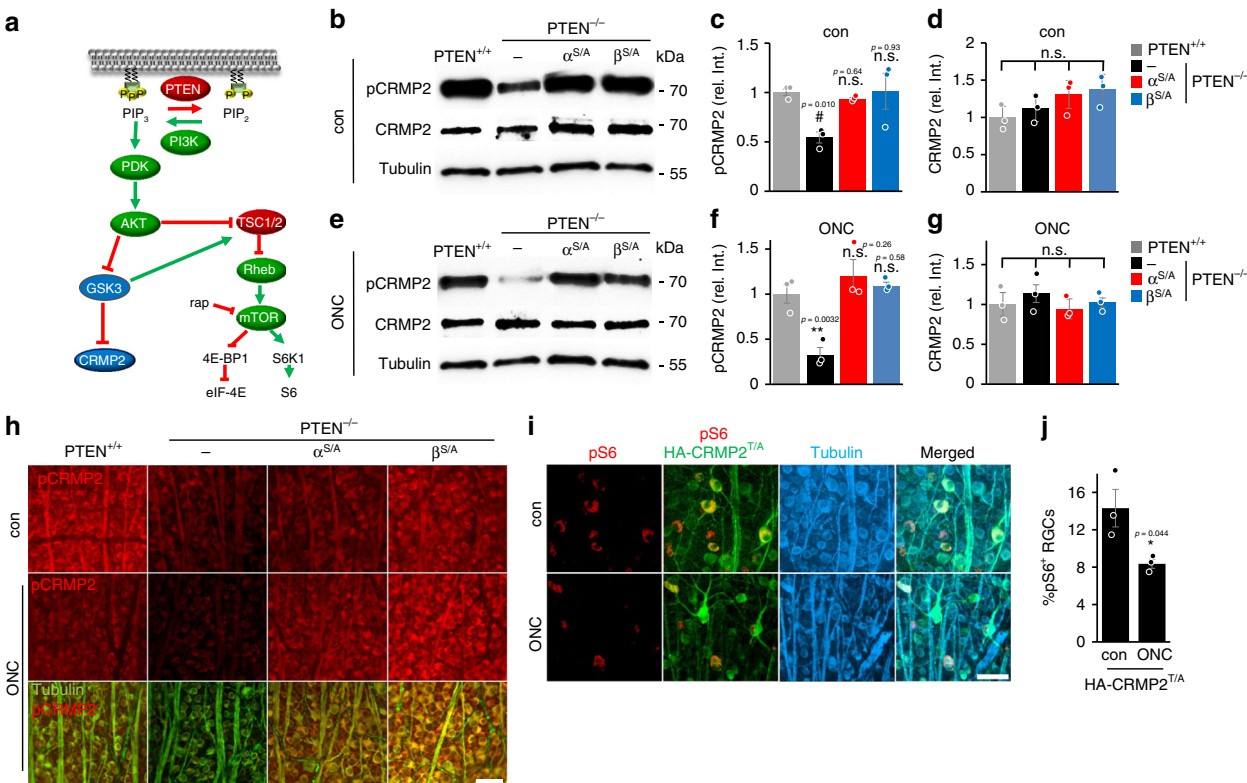

**Fig. 8** PTEN$^{-/-}$ releases CRMP2 from GSK3 inhibition. **a** Schematic illustration of results shown in experiments of (**b–j**). **b, e** Western blots of T514-phosphorylated (pCRMP2) and total CRMP2 in optic nerve lysates from PTEN$^{+/+}$ and PTEN$^{-/-}$ mice without and with *Gsk3α*$^{S/A}$ (α$^{S/A}$), or *Gsk3β*$^{S/A}$ (β$^{S/A}$) knockins. Animals were either left untreated (con, **b**) or subjected to ONC (**e**) 5 days before tissue isolation. Optic nerve segments proximal to the lesion site were used for the analysis. βIII-tubulin (tubulin) was used as a loading control. **c, d, f, g** Densitometric quantification of total CRMP2 and pCRMP2 relative to tubulin and normalized to PTEN$^{+/+}$ on Western blots as depicted in (**b**) and (**e**). Levels of pCRMP2 were considerably compromised upon PTEN$^{-/-}$. Decreased levels were reversed in α$^{S/A}$ and β$^{S/A}$ genotypes, while total CRMP2 remained unaltered in all experimental groups. Values represent means ± SEM of three optic nerves (n = 3) per group. **h** Retinal flat mounts isolated from animals treated as described in (**b**) and (**e**). Retinae were stained for T$^{514}$-pCRMP2 (red) and βIII-tubulin (tubulin, green). ONC and to a stronger extent PTEN$^{-/-}$ reduced pCRMP2 levels in RGC somas and axons compared with the PTEN$^{+/+}$, while levels remained high in PTEN$^{-/-}$/GSK3$^{S/A}$ mice. Scale bar: 50 μm. **i** Retinal flat mounts from PTEN$^{+/+}$ mice either untreated or 5 days upon ONC that were injected with AAV-HA-CRMP2$^{T514/A}$ 3 weeks before the injury. The activity of mTOR was analyzed by pS6 levels (red), while HA-staining (HA-CRMP2$^{T/A}$, green) identified transduced βIII-tubulin-positive RGCs (tubulin, cyan). Scale bar: 25 μm. **j** Quantification of pS6-positive RGCs in retinal flat mounts as described in (**i**). Values represent means ± SEM of three retinae per group (n = 3). Significances of intergroup differences were evaluated using one-way analysis of variance (ANOVA) with Holm-Sidak post hoc test (**c, d, f, g**), or student's *t*-test (**j**). Treatment effects were compared with PTEN$^{+/+}$ con: ##$p < 0.01$, n.s. = non-significant; and PTEN$^{+/+}$ ONC: ***$p < 0.001$, n.s. = non-significant. Treatment effect in (**j**): *$p < 0.05$

similar levels of total CRMP2 protein in all groups (Fig. 8b, d, e, g). Levels of pCRMP2 were significantly reduced in optic nerves of PTEN$^{-/-}$ mice (either untreated or crushed) compared with respective PTEN$^{+/+}$ animals (Fig. 8b, c, e, f). This effect was abolished entirely in animals with *Gsk3α*$^{S/A}$ or *Gsk3β*$^{S/A}$ knockins (Fig. 8b, c, e, f, Supplementary Figure 7a–d). Moreover, immunohistochemical staining of retinal flat mounts from similarly treated animals confirmed the PTEN$^{-/-}$ mediated decrease of pCRMP2 levels in the somata and axons of βIII-tubulin-positive RGCs and the absent effect in both GSK3 knockin genotypes (Fig. 8h).

To test the possibility of whether active CRMP2 also indirectly contributes to the modulation of mTOR activity as was shown for GSK3 inhibition (Fig. 5), we virally expressed constitutively active CRMP2$^{Thr514/Ala}$ (CRMP2$^{T/A}$), which is resistant to inhibitory phosphorylation by GSK3β in RGCs of wt mice[27,33]. These mice remained either untreated or were subjected to ONC. The percentage of pS6-positive cells in CRMP2$^{T/A}$ expressing RGCs (Fig. 8i, j) was very similar to, respectively, treated non-transduced controls (Fig. 5b, c). Moreover, there was no correlation between CRMP2$^{T/A}$ transduction and positive pS6 staining in RGCs. Therefore, in contrast to GSK3β$^{-/-}$, active CRMP2 did not affect mTOR activity.

**Active CRMP2 is essential for PTEN$^{-/-}$-induced axon regeneration.** As PTEN$^{-/-}$ activated CRMP2 in a GSK3-dependent manner and facilitated mTOR activity, we next compared the contribution of both molecules (CRMP2 and mTOR) in PTEN$^{-/-}$-mediated axon regeneration. To this end, we intravitreally injected either AAV-Cre or AAV-GFP in PTEN$^{f/f}$ mice and performed ONC 3 weeks after that. After another 5 days, retinae were used for culture preparation and the regenerative state of RGCs was again assessed by spontaneous neurite growth on a growth permissive (PDL) or inhibitory (CNS myelin) substrate. Cells were cultured in the presence of the CRMP2 inhibitor (R)-lacosamide[34,35] or the mTOR inhibitor rapamycin[4,27] either alone or in combination. Rapamycin-treated groups received

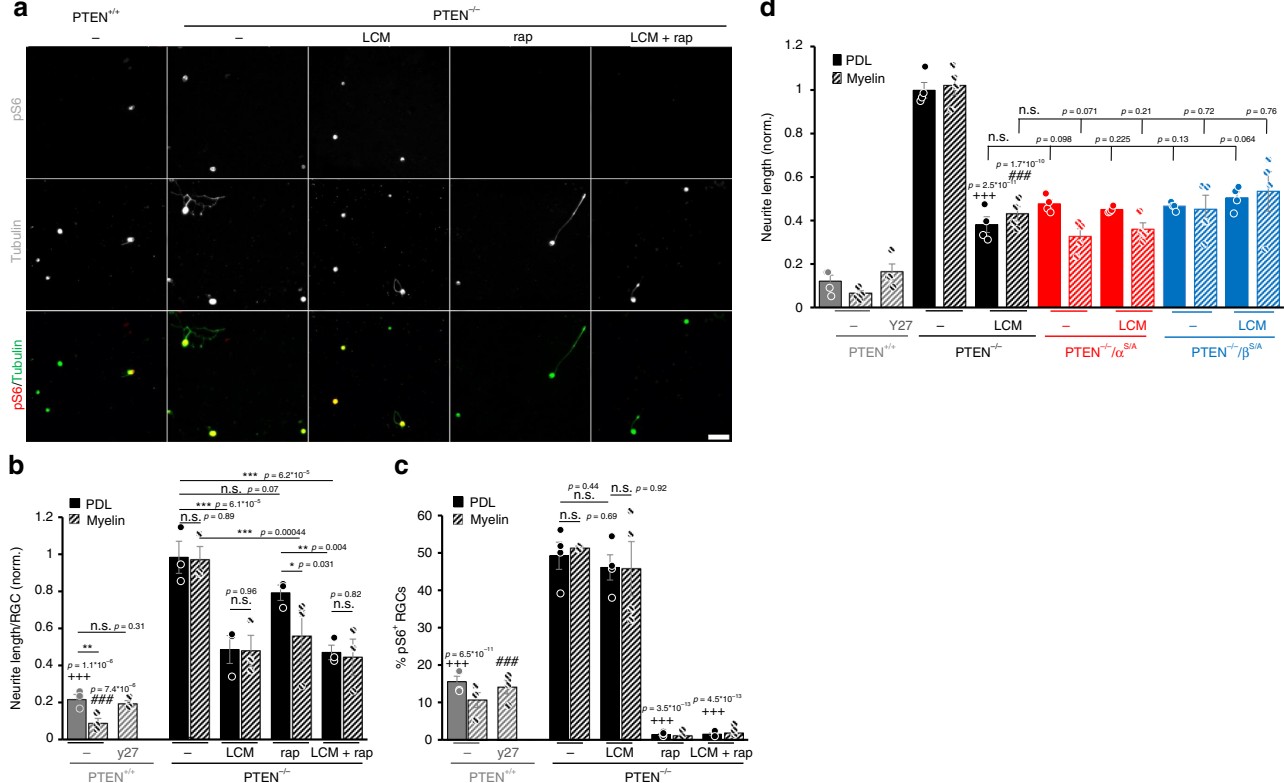

**Fig. 9** PTEN$^{-/-}$ stimulated neurite growth is CRMP2 dependent. **a** Representative pictures of retinal cell cultures plated on PDL. PTEN$^{+/+}$ and PTEN$^{-/-}$ mice were subjected to ONC to transform RGCs into a regenerative state. Cells were cultured 5 days after that. Cultures were either untreated (−) or incubated with rapamycin (rap; 10 nM), lacosamide (LCM, 5 μM), or a combination of both. After 2 days in culture, RGCs were stained for phosphorylated S6 (pS6, red) and βIII-tubulin (tubulin, green). **b** Quantification of neurite growth of RGCs as described in (**a**) in the absence (PDL) or presence of CNS myelin (myelin). Some of the cultures were treated with the ROCK inhibitor Y27632 (Y27, 10 μM). Values represent means ± SEM of three (n = 3) independent experiments with four wells as technical replicate per group and were normalized to the PTEN$^{-/-}$, PDL group with an average neurite length of 6.12 μm per RGC of in total ~500 RGCs with ~30 RGCs showing neurite growth. **c** Quantification of the percentage of pS6-positive RGCs in cultures as described in (**a**). Values represent means ± SEM of 4 independent experiments (n = 4). **d** Quantification of neurite length of βIII-tubulin-positive RGCs from PTEN$^{+/+}$, PTEN$^{-/-}$, or PTEN$^{-/-}$ mice with a Gsk3α$^{S/A}$ (PTEN$^{-/-}$/α$^{S/A}$) or Gsk3β$^{S/A}$ (PTEN$^{-/-}$/β$^{S/A}$) knockin. Animals were subjected to ONC 5 days before culture preparation. Cultures were grown in the in the absence or presence of CNS myelin and either left untreated (−) or incubated with lacosamide (LCM, 5 μM) or Y27632 (Y27, 10 μM) and fixed after 48 h. Values represent means ± SEM of n = 4 independent experiments with four wells as technical replicate per group. Significances of intergroup differences were evaluated using two-way (**b**, **c**) or three-way (**d**) analysis of variance (ANOVA) with Holm-Sidak post hoc test. Treatment effects: compared with PTEN$^{-/-}$, PDL: $^{+++}p < 0.001$; PTEN$^{-/-}$, myelin: $^{###}p < 0.001$; or as indicated: $^{*}p < 0.05$, $^{**}p < 0.01$, $^{***}p < 0.001$, n.s. = non-significant

pretreatment in vivo 12 days before culturing, while others were treated with the vehicle. PTEN$^{+/+}$ (wt) RGCs showed moderate spontaneous neurite growth within 2 days in culture (Fig. 9a, b). Neurite growth was significantly reduced in the presence of myelin and rescued by treatment with the ROCK inhibitor Y27632, which reportedly overcomes myelin inhibition[36–38] and thereby verified the specificity of the myelin effect (Fig. 9a, b). As expected, PTEN$^{-/-}$ significantly increased neurite growth and conferred complete disinhibition toward myelin (Fig. 9a, b) accompanied by increased pS6 levels (Fig. 9c). Significantly, LCM markedly decreased PTEN$^{-/-}$-mediated neurite growth on both PDL and myelin without affecting mTOR activity (indicated by pS6 staining) (Fig. 9b, c). In contrast, rapamycin, which abrogated S6 phosphorylation almost completely, showed no significant reduction in neurite growth on PDL and only impairment of disinhibition toward myelin (Fig. 9a–c). Interestingly, the combination of LCM and rapamycin did not further affect neurite growth on PDL or myelin compared with LCM treatment alone. To additionally analyze the contribution of CRMP2 activity in this context, we compared retinal cultures of wt, PTEN$^{-/-}$, or PTEN$^{-/-}$/GSK3$^{S/A}$

mutants in the presence or absence of LCM using a similar experimental setting. All PTEN$^{-/-}$/GSK3$^{S/A}$ genotypes (PTEN$^{-/-}$/ GSK3α$^{S/A}$, or PTEN$^{-/-}$/GSK3β$^{S/A}$) showed similarly reduced neurite growth as LCM treated PTEN$^{-/-}$ cultures on PDL and myelin (Fig. 9d). Moreover, LCM application did not additionally affect neurite growth in PTEN$^{-/-}$/GSK3$^{S/A}$ RGCs, suggesting that CRMP2 inhibition by the GSK3 knockin is already sufficient to achieve maximal inhibitory effects.

**Active CRMP2 rescues adverse GSK3$^{S/A}$ effect on axon regeneration.** Finally, we tested whether virally induced expression of constitutively active CRMP2 (CRMP2$^{T/A}$) could rescue the inhibitory effect of GSK3$^{S/A}$ on PTEN$^{-/-}$-mediated axon regeneration in vivo (Fig. 3). Because CRMP2 phosphorylation (Fig. 8b, c, e, f, h) and axon growth inhibition (Fig. 3a–i) was already maximal with the Gsk3β$^{S/A}$ knockin, we virally expressed constitutively active CRMP2$^{T/A}$ in PTEN$^{-/-}$ and PTEN$^{-/-}$/GSK3β$^{S/A}$ mice. Intravitreally injected AAV-CRMP2$^{T/A}$ mixed with AAV-Cre transduced >70% (CRMP2$^{T/A}$) and >90% (Cre) of RGCs, respectively, with a nearly 100% overlap of CRMP2$^{T/A}$ and Cre-co-expression after

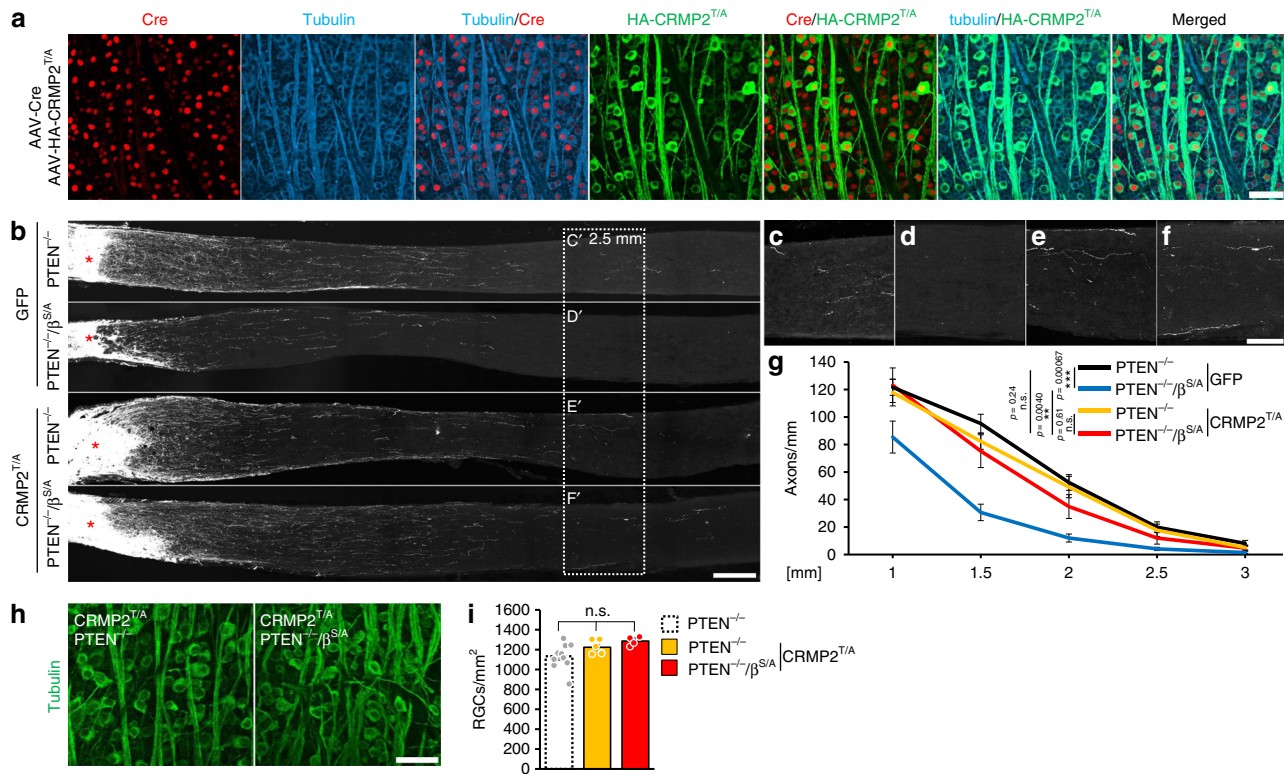

**Fig. 10** CRMP2$^{T/A}$ overcomes GSK3β$^{S/A}$-mediated inhibition on nerve regeneration. **a** Retinal flat-mounts from PTEN$^{-/-}$ mouse 3 weeks after intravitreal injection of AAV-Cre + AAV-HA-CRMP2$^{T/A}$, immunohistochemically stained for Cre (red), the HA-tag of CRMP2$^{T/A}$ (HA-CRMP2$^{T/A}$, green), and βIII-tubulin (tubulin, blue). Scale bar: 50 μm. **b** Representative longitudinal optic nerve sections with regenerating, CTB-labeled axons 3 weeks after ONC. PTEN$^{-/-}$ or PTEN$^{-/-}$/GSK3β$^{S/A}$ mice had received treatment with either AAV-GFP (GFP) or AAV-HA-CRMP2$^{T/A}$ (CRMP2$^{T/A}$) 3 weeks before ONC. Asterisks indicate the lesion site. Scale bar: 200 μm. **c–f** Magnifications of optic nerve sections as indicated in (**b**) at 2.5 mm beyond the lesion site. Scale bar: 100 μm. **g** Quantification of regenerating axons at indicated distances beyond the crush site in optic nerves as described in (**b**). CRMP2$^{T/A}$ rescued adverse effect of Gsk3β$^{S/A}$ knockin on PTEN$^{-/-}$-induced optic nerve regeneration. Values represent means ± SEM of at least seven animals per group (PTEN$^{-/-}$/GFP, $n = 13$; PTEN$^{-/-}$/β$^{S/A}$/GFP, $n = 7$; PTEN$^{-/-}$/CRMP2$^{T/A}$, $n = 12$; PTEN$^{-/-}$/β$^{S/A}$/CRMP2$^{T/A}$, $n = 8$). **h** βIII-tubulin-stained whole-mount retinae from AAV-HA-CRMP2$^{T/A}$ treated PTEN$^{-/-}$ and PTEN$^{-/-}$/GSK3β$^{S/A}$ mice as described in (**b**). Scale bar: 50 μm. **i** Quantification of RGCs per mm$^2$ in retinal whole-mounts as described in (**h**). The number of surviving RGCs 21 days after ONC were similar for the respective genotypes. The dashed bar represents values from PTEN$^{-/-}$ mice as shown in Fig. 4b for comparison. All values represent means ± SEM of 4–10 retinae per group (PTEN$^{-/-}$, $n = 10$; PTEN$^{-/-}$/CRMP2$^{T/A}$, $n = 6$; PTEN$^{-/-}$/β$^{S/A}$/CRMP2$^{T/A}$, $n = 4$). Significances of intergroup differences were evaluated using two-way (**g**) or one-way (**i**) analysis of variance (ANOVA) followed by Holm-Sidak post hoc test. Treatment effects: **$p < 0.01$, ***$p < 0.001$, n.s. = non-significant

3 weeks (Fig. 10a). Respective control groups received AAV-GFP instead or hemagglutinin-tagged constitutively active CRMP2 (HA-CRMP2$^{T/A}$). Consistent with the hypothesis that the release of CRMP2 from GSK3-mediated inhibition is significantly involved in the mechanism, viral overexpression of constitutively active CRMP2$^{T/A}$ did not additionally enhance PTEN$^{-/-}$-induced optic nerve regeneration (Fig. 10b, c, e, g), but rescued the negative effect of GSK3β$^{S/A}$ (Fig. 3h, Fig. 10b–g, Supplementary Fig. 8a–d) 3 weeks after ONC. CRMP2$^{T/A}$ expression did not affect the survival of RGCs with PTEN$^{-/-}$ or PTEN$^{-/-}$/GSK3β$^{S/A}$ (Fig. 10h, i).

## Discussion

PTEN$^{-/-}$-mediated neuroprotection and axon regeneration have been mainly linked to the PI3K/AKT mediated prevention of mTOR inhibition in neurons upon axotomy[4,7]. While constitutively active AKT is as capable as PTEN$^{-/-}$[9,10], recent data demonstrate that direct activation of mTOR by constitutively active Rheb or downstream modulation of the mTORC1 effectors E4BP or S6K1 fail to fully mimic the regenerative effect[15,17,18,39]. These findings raise the question of which molecular actors downstream of AKT are involved in mediating axon extension? Taking advantage of genetic mouse models, the current study

shows that PTEN$^{-/-}$ releases axonal CRMP2 from GSK3-mediated inhibition and that this mechanism substantially contributes to the axon regeneration promoting effect. Moreover, we demonstrate that PTEN$^{-/-}$ also partially activates mTOR indirectly via its inhibitory effect on GSK3. These mechanistic insights may help to find alternative strategies to mimic a robust regenerative stimulus without the high cancerogenic risk associated with PTEN depletion.

The inactivation of GSK3 and subsequent release of axonal CRMP2 from inhibition is an essential part of the mechanism underlying the axon growth promoting effect of PTEN$^{-/-}$. This is supported by the findings that PTEN$^{-/-}$ markedly increased inhibitory phosphorylation of both GSK3 isoforms and reduced CRMP2 phosphorylation in axons. When these effects were blocked entirely in PTEN$^{-/-}$/GSK3$^{S/A}$ mice with high GSK3 activity, PTEN$^{-/-}$ stimulated axon regeneration in vivo, as well as neurite growth in culture, were significantly compromised. These negative effects on regeneration were mimicked by the CRMP2 inhibitor LCM and, on the other hand, overcome by exogenous expression of constitutively active CRMP2 in RGCs, which was specifically resistant to inhibitory GSK3 mediated phosphorylation at T514.

Moreover, we found that PTEN$^{-/-}$ triggered mTOR activation partially occurred indirectly via its inhibitory effect on GSK3. Consistently, higher GSK3 activity significantly reduced the induction of mTOR activity in PTEN$^{-/-}$/GSK3$^{S/A}$-RGCs compared with PTEN$^{-/-}$-RGCs, while the depletion of GSK3β (GSK3β$^{-/-}$) and even more effectively total GSK3(α/β)$^{-/-}$ in PTEN$^{+/+}$ mice showed the opposite effect. This GSK3-dependent modulation of S6 phosphorylation was entirely blocked by rapamycin, indicating that it must have been regulated upstream of mTOR. Although modulation of mTOR activity by GSK3 has not yet been described in neurons, in non-neuronal cells active GSK3 was shown to phosphorylate and activate TSC2, which subsequently leads to Rheb and mTOR inhibition[31]. Whether a similar mechanism underlies the effects in RGCs (neurons) needs to be investigated in the future. On the other hand, rapamycin treatment did not affect PTEN$^{-/-}$-mediated phosphorylation of GSK3α or GSK3β, thereby excluding a potential feedback loop of S6K1 on both isoforms in RGCs as previously shown in mouse embryonic fibroblasts[40]. Although the underlying mechanism awaits further investigation, the identified regulatory effect of GSK3 on mTOR activity in RGCs contradicts a previous study, suggesting that GSK3β$^{-/-}$ does not affect pS6 levels in RGCs and that both substrates act entirely independently from each other[10]. The differing findings of both studies could be because Guo et al. quantified pS6 staining intensities of RGCs on selected retinal sections, whereas we quantified RGCs in entire retinal whole-mounts. In addition, we used Western blot analysis to verify our findings and contrary to Guo et al., our quantification method in retinal whole-mounts showed a reduction of pS6 levels in RGCs upon ONC, which is consistent with the existing literature[4,32]. Therefore, full PTEN$^{-/-}$-mediated activation of mTOR occurs to some extent indirectly via its inhibitory effect on GSK3 and both AKT targets cannot be regarded as independent from each other. In fact, the incomplete mTOR activation in PTEN$^{-/-}$/GSK3$^{S/A}$ mice may also be the reason why the overexpression of CRMP2$^{T/A}$ only approximately rescued these animals from reduced optic nerve regeneration compared with PTEN$^{-/-}$ alone (Fig. 10b–g) and thereby reflected the contribution of mTOR to regeneration, for instance by maintaining neurons in the regenerative state as shown previously[32].

Knockout of GSK3α did not measurably alter mTOR activity in non-injured RGCs but prevented its inactivation upon axotomy. As GSK3α$^{-/-}$ did neither promote optic nerve regeneration nor neuroprotection[27] these findings suggest that just maintaining mTOR activity at its levels in naive RGCs after axotomy is insufficient to achieve beneficial effects. In contrast to GSK3α, GSK3β$^{-/-}$ increased levels of active CRMP2[27] and, additionally slightly increased mTOR activity in non-injured and axotomized RGCs. Although this treatment did not lead to any neuroprotection of injured RGCs, it resulted in moderate regeneration of axons in the optic nerve[27]. As mTOR inhibition by rapamycin does not affect GSK3β$^{-/-}$-mediated optic nerve regeneration[10], these data suggest that the effect on regeneration was mostly due to the release of active endogenous CRMP2 by GSK3β$^{-/-}$ rather than the impact on mTOR[27]. Moreover, the observation that GSK3β$^{-/-}$ failed to rescue axotomized RGCs from degeneration and did not increase soma size, while both PTEN$^{-/-}$ and the GSK3(α/β)$^{-/-}$ induced higher pS6 levels, soma sizes and were neuroprotective suggests that mTOR only above a certain activity level affects neuroprotection and soma size. Consistently, only the double knockin Gsk3 (α/β)$^{S/A}$, but not the single Gsk3β$^{S/A}$ knockin compromised PTEN$^{-/-}$ dependent soma size, which was previously described as a consequence of S6k1 activity downstream of mTOR[15]. Also, the higher pS6 levels and soma sizes observed after PTEN$^{-/-}$ compared with GSK3(α/β)$^{-/-}$ correlated with stronger neuroprotection and may be due to the additional mTOR activation via the PI3K/AKT/Rheb axis.

While mTOR activity seems to be mostly responsible for preventing apoptotic cell death after PTEN$^{-/-}$, the release of active CRMP2 appears to be essential for axon regeneration. In support of this idea, TSC1$^{-/-}$ or expression of constitutively active Rheb or S6K1 reportedly mimicked PTEN$^{-/-}$-mediated survival of axotomized RGCs, but fail to promote substantial optic nerve regeneration[4,15,17]. Consistently, we found that rapamycin, although completely preventing S6 phosphorylation, did not reduce PTEN$^{-/-}$-induced neurite growth of RGCs cultured on a growth permissive substrate and had only a small effect on disinhibition toward CNS myelin (Fig. 9b, c). In contrast, the CRMP2 inhibitor LCM[27,34,35] significantly compromised PTEN$^{-/-}$-induced neurite growth on permissive and inhibitory substrates (myelin) and was not further affected by additional rapamycin. Moreover, the inhibitory effect of LCM in PTEN$^{-/-}$ was similarly strong as observed in Pten$^{-/-}$/Gsk3$^{S/A}$ knockin mice where lacosamide did not further compromise neurite elongation of RGCs from these animals. Most importantly, expression of constitutively active CRMP2$^{T/A}$ almost entirely rescued impaired optic nerve regeneration in PTEN$^{-/-}$/GSK3β$^{S/A}$ animals despite reduced mTOR activity (Figs. 5, 10), underlining its role in this context.

Whether inhibition of the remaining mTOR activity reduces the regenerative outcome in the presence of active CRMP2 was not addressed in the current study. However, as stated above, it was previously demonstrated that rapamycin does not reduce GSK3β$^{-/-}$-mediated axon regeneration[10]. As GSK3β$^{-/-}$ leads to CRMP2 and mTOR activation[27], it is quite possible that the remaining mTOR activity is not required for the CRMP2$^{T/A}$ effect on regeneration. Moreover, CRMP2$^{T/A}$ did not additionally enhance regeneration in PTEN$^{-/-}$ mice suggesting that CRMP2 is already sufficiently activated by PTEN$^{-/-}$. Thus, these data obtained from cell culture and in vivo experiments substantiate the involvement of the GSK3/CRMP2 pathway in the PTEN$^{-/-}$-mediated effect on axon regeneration without excluding any contribution of other possible, PI3K/AKT independent mechanisms, such as an influence on the activity of focal adhesion kinase (FAK), RAS or extracellular signal-regulated kinase (ERK)[41,42]. Whether co-expression of constitutively active CRMP2 and mTOR activation, for example by constitutively active Rheb, can fully mimic the effect of PTEN needs to be addressed in the future. Nevertheless, activation of downstream targets without affecting PTEN itself is, from a clinical point of view, more feasible, because PTEN inhibition is highly cancerogenic[43]. Therefore, the finding of a role of the GSK3/CRMP2 pathway in PTEN$^{-/-}$-mediated axon growth may open new therapeutic strategies for CNS repair.

## Methods

**Mouse strains**. Male and female mice (2–3 months old) were used for all experiments: GSK3β$^{Ser9Ala}$, GSK3α$^{Ser21Ala}$, and GSK3α$^{Ser21Ala}$/β$^{Ser9Ala}$ double knockin mice (all C57BL/6,129/Ola background) and respective wt C57BL/6,129/Ola mice[29] were provided by Prof. Dr. Dario Alessi (University of Dundee, UK) and crossbred with PTEN$^{f/f}$ mice (C57BL/6;129) provided by Prof. Dr. Raab (University of Düsseldorf, Germany). GSK3β$^{f/f}$ and GSK3α$^{f/f}$ mice (C57BL/6;129) were provided by Prof. Dr. Woodgett (University of Toronto, Canada)[27]. For all experiments, RGC-specific Pten or Gsk3 knockout was induced by intravitreal injection of AAV-Cre into the respective floxed mice (PTEN$^{f/f}$ and GSK3α, β, (α/β)$^{f/f}$) 3 weeks before optic nerve injury or tissue collection as described previously with transduction rates of >90%. All mice were housed under the same conditions for at least 10 days before experiments and generally maintained on a 12-h light/dark cycle with ad libitum access to food and water. All experimental procedures were performed in compliance with ethical regulations and approved by the local animal care committee (LANUV Recklinghausen) and conducted in compliance with federal and state guidelines for animal experiments in Germany.

**Surgical procedures**. Depletion of PTEN, GSK3α and/or GSK3β was induced by intravitreal injection of 2 µl AAV-Cre into the, respectively, floxed mice 3 weeks before surgery. CRMP2 lacking the phosphorylation site at T$^{514}$ (CRMP2$^{T/A}$) was overexpressed upon intravitreal injection of AAV-CRMP2$^{T/A}$. In all cases, respective control animals were intravitreally injected with AAV-GFP. For surgery, animals were anesthetized by intraperitoneal injections of ketamine (120 mg/kg) and xylazine (16 mg/kg). The left optic nerve was intraorbitally crushed 1 mm behind the eyeball for 10 s using jeweler's forceps (Hermle, Tuttlingen, Germany), as described previously[22]. Two days before tissue isolation, regenerating axons were labeled by intravitreal injection of 2 µl Alexa Flour 594- or Alexa Flour 488-conjugated cholera toxin β subunit (CTB, 0.5% in PBS; Molecular Probes, Carlsbad, Eugene, USA).

The specific mTOR inhibitor rapamycin (5 mg/kg, LC Laboratories) or the vehicle (5% Ethanol, 5% Tween 80, 5% polyethylene glycol 400 in PBS) was injected intraperitoneally starting 1 week before surgery. Rapamycin was dissolved at 20 mg/ml in ethanol and diluted in 5% Tween 80, 5% polyethylene glycol 400 in PBS before each experiment, and administered every 2 days[4,32].

**Western blot assays**. For protein lysate preparation, mice were killed, and retinae or optic nerve segments proximal to the lesion site were isolated and dissected. Respective tissues were homogenized in lysis buffer (20 mM Tris/HCl pH 7.5, 10 mM KCl, 250 mM sucrose, 10 mM NaF, 1 mM DTT, 0.1 mM Na$_3$VO$_4$, 1% Triton X-100, 0.1% SDS) with protease inhibitors (Calbiochem, Darmstadt, Germany) and phosphatase inhibitors (Roche, Basel, Switzerland) using 5 sonication pulses at 40% power (Bandelin Sonoplus, Berlin, Germany). Lysates were cleared by centrifugation in an Eppendorf tabletop centrifuge at $4150 \times g$ for 10 min at 4 °C. Proteins were separated by sodium-dodecyl-sulfate-polyacrylamide gel electrophoresis (SDS-PAGE), using Mini TGX gels (10%, Bio-Rad, Hercules, USA) according to standard protocols and transferred to nitrocellulose membranes (0.2 µm, Bio-Rad, Hercules, USA). Blots were blocked in 5% dried milk in phosphate-buffered saline with 0.05% Tween-20 (PBS-T) (Sigma, St Louis, USA) and incubated with antibodies against βIII-tubulin (1:2000; BioLegend, San Diego, USA, RRID:AB_2313773), S9-phosphorylated GSK3β (1:1000, RRID:AB_2115196) S21-phosphorylated GSK3α (1:2000; RRID: AB_2114897), total GSK3α/β (1:500, RRID:AB_10547140; all from Cell Signaling Technologies, Cambridge, UK), ribosomal protein S6 (1:5000 Cell Signaling Technologies RRID:AB_2181035), T$^{514}$-phosphorylated CRMP2 (1:2000; Abcam, Cambridge, UK, RRID:AB_942229), total CRMP2 (1:500; Cell Signaling Technologies, RRID:AB_2094339), or T308-phosphorylated AKT (1:1000; Cell Signaling Technologies, RRID:AB_2629447) at 4 °C overnight. All antibodies were diluted in PBS-T containing 5% bovine serum albumin (BSA, Sigma). Anti-rabbit or anti-mouse immunoglobulin G (IgG) conjugated to horseradish per-oxidase (1:80,000; Sigma) were used as secondary antibodies. Phospho-GSK3α and CRMP2 primary antibodies were detected using conformation-specific horseradish peroxidase conjugated anti-rabbit (IgG) secondary antibody (1:4000, Cell Signaling Technologies). Antigen-antibody complexes were visualized using an enhanced chemiluminescence substrate (Bio-Rad) on a FluorChem E detection system (ProteinSimple, San Jose, USA). Western blots were repeated at least three times to verify results. Band intensities were quantified relative to respective loading controls using ImageJ software.

**Immunohistochemistry**. Animals were anesthetized and intracardially perfused with cold saline followed by PBS containing 4% paraformaldehyde (PFA). Eyes with or without attached optic nerves were removed from connective tissue, post-fixed overnight in 4% PFA solution at 4 °C and subsequently transferred to 30% sucrose for at least 4 h. The tissue was then embedded in KP-cryo com-pound (Klinipath, Duiven, Holland) and longitudinal sections (14 µm) were cut on a cryostat, thaw-mounted onto charged glass slides (Superfrost Plus, VWR, Darmstadt, Germany) and stored at −20 °C until further use.

Retinal whole-mounts were prepared, fixed in 4% PFA for 30 min at room temperature and incubated in PBS containing 2% Triton X-100 for 1 h to improve antibody penetration. RGCs in retinal whole-mounts and sections were visualized using a monoclonal antibody against βIII-tubulin (1:1000; BioLegend, RRID: AB_2313773).

Whole-mounts and sections were immunohistochemically stained with antibodies against S9-phosphorylated GSK3β (1:1000, RRID:AB_2115196) S21-phosphorylated GSK3α (1:2000; RRID:AB_2114897) and total GSK3α/β (1:500, RRID:AB_10547140; all from Cell Signaling Technologies, Cambridge, UK). Transduction with AAV-Cre was verified by GFP co-expression or staining of the HA-tagged Cre-recombinase using a polyclonal HA antibody (1:500; Sigma, St Louis, USA, RRID:AB_260070). GFP expression was amplified by staining with a polyclonal antibody (1:500, Novus Biologicals, Minneapolis, USA, RRID: AB_10128178). T$^{514}$-phosphorylated CRMP2 and phosphorylated ribosomal protein S6 were visualized using polyclonal antibodies (1:2000; Abcam, RRID: AB_942229; 1:500 Cell Signaling Technologies RRID:AB_2181035). Secondary antibodies (all 1:1000) included donkey-anti-mouse, anti-goat and anti-rabbit IgG antibodies conjugated to Alexa Fluor 488, 594 or 350 (Molecular Probes, Carlsbad, USA). Sections and whole-mounts were cover-slipped with Mowiol and analyzed

using either a fluorescent (Axio Observer.D1; Zeiss, Jena, Germany) or confocal laser scanning microscope (LSM 510, Zeiss or SP8, Leica).

**Quantification of axons in the optic nerve**. As described before[44], regeneration of axons was quantified over the entire length of the optic nerve and chiasm for all animals as previously described[30,32]. In brief, eyes were isolated with optic nerves attached and prepared for histology 21 days after surgery as described above for immunohistochemistry. The number of CTB-labeled axons extending 0.5,1, 1.5, 2, 2.5, 3, and 3.5 mm from the injury site was quantified on at least six nerve sections per animal. For this purpose, pictures of optic nerve sections were taken under ×200 magnification using a fluorescent microscope (Axio Observer.D1, Zeiss). Subsequently, axons were counted and normalized to the cross-sectional width of the optic nerve at the respective measuring points. Each experimental group included 6–13 mice.

**Quantification of RGCs in retinal whole-mounts**. As described previously[45], for quantification of surviving RGCs, retinal whole-mounts were stained with an antibody against βIII-tubulin (1:1000; BioLegend, RRID:AB_2313773). Retinae were divided into four quadrants. In each quadrant, 4–5 non-overlapping pictures were taken using a fluorescent microscope (×400, Axio Observer.D1, Zeiss), pro-ceeding from the center to the periphery. The average number of βIII-tubulin-positive RGCs per picture was determined and normalized to an area of 1 mm². Values were averaged per retina and then across all similarly treated animals to obtain a group mean ± SEM. Six to ten retinae were analyzed per experimental group.

**Preparation of AAV2**. For recombinant AAV2 (afterwards referred to as AAV) production, pAAV-MCS plasmid (Stratagene, La Jolla, USA) carrying either cDNA for Cre-HA (kindly provided by Prof. Dr. Zhigang He, Harvard Medical School, Boston, USA)[4], CRMP2$^{Thr514/Ala}$, or GFP downstream of the CMV promoter was used. AAV-293 cells (Stratagene) were co-transfected with the respective pAAV-MCS plasmid, pAAV-RC (Stratagene) encoding the AAV rep and cap genes and the helper plasmid pHGTI-Adeno1 encoding E24, E4, and VA. Purification of virus particles was performed as described previously[4,46]. Mainly RGCs are transduced upon intravitreal injection of AAV2[47,48] as this virus serotype is highly neurotropic[4,49] and RGCs are the first neurons to be encountered by the virus.

**Dissociated retinal cell cultures**. As described previously[27], mice were sub-jected to ONC in vivo. The specific mTOR inhibitor rapamycin (5 mg/kg, LC Laboratories) or the vehicle (5% Ethanol, 5% Tween 80, 5% polyethylene glycol 400 in PBS) was injected intraperitoneally starting 1 week before surgery. Five days after ONC, retinal cultures were prepared as described previously[50]. In brief, tissue culture plates (4-well plates; Nunc, Wiesbaden, Germany) were coated with poly-D-lysine (PDL, 0.1 µg/ml, molecular weight between 70,000 and 150,000 Da; Sigma, St Louis, USA) for 1 h at room temperature, rinsed three times with distilled water and air-dried. Some of the culture plates were addi-tionally coated with inhibitory CNS myelin extract (200 µl/well), obtained as described previously[36]. Myelin was applied at a preoptimized concentration of ~12.5 µg/ml and dried overnight. To prepare low-density retinal cell cultures, mice were killed by cervical dislocation. Retinae were rapidly dissected from the eyecups and incubated at 37 °C for 30 min in a digestion solution containing papain (10 U/mL; Worthington) and L-cysteine (0.2 µg/mL; Sigma) in Dul-becco's modified Eagle medium (DMEM; Thermo Fisher). Retinae were then rinsed with DMEM and triturated in 2.5 ml DMEM containing B27-supplement (1:50; Invitrogen; Carlsbad, USA) and penicillin/streptomycin (1:50; Biochrom, Berlin, Germany). Dissociated cells were passed through a cell strainer (40 µm, BD Falcon, Franklin Lakes, USA) and 300 µl of cell suspension was added into each well. Rapamycin dissolved in DMSO was applied into cultures at 10 nM. The ROCK inhibitor Y27632 (Sigma) was added at a concentration of 10 µM to verify the specificity of growth-inhibitory myelin as described previously[30,32]. The previously characterized CRMP2 inhibitor (R)-lacosamide (LCM[35], Biozol) was used at final concentrations of 5 µM. Retinal cells were cultured for 48 h and then fixed with 4% PFA (Sigma).

After fixation with 4% PFA for 30 min at RT, cell cultures were processed for immunocytochemical staining with a βIII-tubulin-antibody (1:2000, BioLegend, RRID:AB_2313773). All RGCs with regenerated neurites were photographed using a fluorescent microscope (×200, Axio Observer.D1, Zeiss) and neurite length was determined using ImageJ software. Also, the total number of βIII-tubulin-positive RGCs with an intact nucleus (6-diamidino-2-phenylindole (DAPI)) per well was quantified to test for potential neurotoxic effects. The average neurite length per RGC was determined by dividing the sum of neurite length per well by the total number of RGCs per well (on average ~500 RGCs). The resulting neurite length per RGC also includes the majority of RGCs which did not grow out any neurite (~90%) thereby representing neurite growth as well as neuritogenesis. Thus, obtained values are much lower than the average axon length of RGCs with extended neurites (~20–200 µm). Cultures were arranged in a pseudo-randomized manner on the plates so that the investigator would not be aware of their identity.

Data represent means ± SEM of at least three independent experiments each with four replicate wells per treatment.

**Statistical analysis**. Significances of intergroup differences were evaluated using Student's *t*-test or analysis of variance (ANOVA) followed by Holm-Sidak or Tukey post hoc test using GraphPad Prism software.

**Reporting summary**. Further information on research design is available in the Nature Research Reporting Summary linked to this article.

## Data availability
All data generated or analyzed during this study are included in this published article (and its supplementary information files).

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

## Acknowledgements

We thank Marcel Kohlhaas and Rebecca Schnepf for technical support. This work was supported by the German Research Foundation.

## Author contributions

D.F. designed concept and supervised research; M.L., A.H., A.A. and D.F. performed research; M.L., A.A., A.H. and D.F. analyzed data; and M.L. and D.F. wrote the paper.

## Additional information

**Competing interests:** The authors declare no competing interests.

