## [Peer Review File · Communications Biology]

Reviewers' comments:

Reviewer #1 (Remarks to the Author):

The manuscript by Leibinger et al. investigates the signaling principles relevant to PTEN-loss mediated growth and regeneration responses using the optic nerve as model system. Work by others identified (a while back) robust installment of axon growth following a crush injury to the optic nerve in PTEN deficient retinal ganglion cells. This response was shown to involve PI3K/PTEN dependent regulation of downstream mTOR function. On the other hand, this current group previously demonstrated that axon regeneration in the same injury model can be modulated by the PI3K/PTEN and Akt regulated kinase GSK3.

Here, the group uses a number of different experimental system and combinational approaches to analyze the possibility that GSK3 (as well as CRMP2) is involved in mediating PTEN-loss induced axon regeneration responses. The main conclusion is that, indeed, GSK3/CRMP2 is essential for PTEN-/- mediated axon regeneration. The authors employ a number of epistasis experiments to detail the signalling relationship. For example, they show that maintenance of GSK3 activity in mice with overactivated GSK3 compromised PTEN-loss mediated optic nerve regeneration. One of the major findings of this study is that GSK3 in itself is able to activation mTOR function in crush injury models, indicating that one could bypass PI3K/PTEN to modulate this pathway. The work is well executed and results are convincing. However, the paper is difficult to read, not alone because of the difficult signaling relationships tested. The paper is an obvious progression from previous publications, and therefore deserves publication. I only have few comments on how to improve the work.

It would be of interest to see if the increase in GSK3 phosphorylation in control animals following ONC dependent on Rapamycin?

A further readout for PTEN-loss is the increase in neuronal soma size. The data indicate that there might be a dichotomy on this PI3K/PTEN dependent function, which would be of great interest to the community. The authors should analyze cell size of RGC in the presence/absence of PTEN with or without constitutive activity of GSK3alpha/beta (see Figure 4).

The discussion is stating the obvious that is already introduced/mentioned in the introduction/result sections. It would be of great interest to incorporate further topics, such as the potential involvement of PTEN dependent, but PI3K independent functions. Do the authors believe that all regenerative growth that can be installed through PTEN-loss involve its lipid phosphatase (i.e. is all PTEN-loss induced growth dependent on mTOR and GSK3)? Or is it conceivable that PTEN dependent, but PI3K independent function could regulate axon regeneration.... Or that PTEN feedback regulation by GSK3 (PTEN T366) is involved in axon regeneration? I believe the authors could expand discussing the implications of their findings further....

Reviewer #2 (Remarks to the Author):

Phosphatase and tensin homolog (PTEN) has been identified as a prominent intrinsic inhibitor of CNS axon regeneration. PTEN knockout mediates axon regeneration in CNS through the PI3K/AKT/TSC signaling pathway and mTOR is a key downstream signal of PI3K/AKT in regulating the process of cellular growth. However, The precise mechanisms by which PTEN/mTOR controls axon regeneration remain to be elucidated. in this study, Leibinger et al. utilized different genetic mouse models to investigate the underlying mechanism of knockout of (PTEN) promoting axon regeneration after optic

nerve crush injury. Their study shows that PTEN knockout significantly increased inhibitory phosphorylation of both GSK3 α and GSK3 β and reduced CRMP2 phosphorylation. The inactivation of GSK3 and the downstream release of axonal CRMP2 from inhibition is an essential part of the mechanism of the axon growth promoting effect of PTEN knockout. PTEN knockout induces inhibitory GSK3 phosphorylation is mTOR independent. Although this hypothesis needs to be further confirmed in other more common CNS injury model such as spinal cord injury, targeting GSK3/CRMP2 pathway could be a new strategy to promote axon regeneration in CNS in order to avoid the potential risk of cancer triggering by direct inhibition of PTEN.

Authors applied β III-tubulin as a marker for RGC in their histology study and RGC survival quantification. β III-tubulin is a general neuronal marker but not a specific RGC marker. RBPM5 and/or Brn3a should be the selective markers of ganglion cells in the mammalian retina.

Type error in Line 610. I should be H instead.

Reviewer #3 (Remarks to the Author):

The manuscript by Leibinger et al. reported a detailed signaling pathway downstream of Pten knockout in promoting optic nerve regeneration. Specifically, the study showed that downstream of Pten inhibition, GSK3 inactivation and the subsequent activation of CRMP2 were crucial for optic nerve regeneration. In particular, direct activation of CRMP2 was able to fully rescue Pten-deletion induced optic nerve regeneration without GSK3 phosphorylation and mTor activation. Overall, the results presented in the study were largely nice and clear. The major strength of the study is the identification of CRMP2 as an important downstream mediator of Pten-GSK3 pathway to regulate optic nerve regeneration. The major weakness was the lack of overall novelty. Addressing the following comments with either new experiments or text revising would greatly strengthen the manuscript.

1. In Figure 2, all the results were based on western blot of the whole retina tissue, in which proteins from RGCs were only a very small percentage of the total retina proteins. Thus, the results were not convincing enough. The authors should at least mention the potential weakness.

2. In Figure 3, the results were very similar to a previous published study, in which active Akt-induced optic nerve regeneration was impaired by GSK3 β -S9A or GSK3 α -S21A mutants. In Figure 3A, B, the axon lengths in many conditions were shorter than 4 μ m, probably less than the cell diameter. It is difficult to accurately quantify axon length with such short axons.

3. In Figure 4, It showed that activation of GSK3 α /b had no negative effects on RGC survival promoted by Pten knockout after nerve crush. This result suggests that GSK3 activation have no negative effect on Pten knockout-induced Akt-mTor activation. In Figure 5B, C, Pten knockout-induced mTor activation (increased level of pS6) was reduced by GSK3 activation (SA mutants). In Figure 5I, J double knocking out GSK3 α /b increased RGCs survival by activation of mTor. Taken together, it seems that GSK3 activation reduces Pten knockout-induced mTor activation but has no negative effect on neuronal survival, whereas GSK3 knockout activates mTor and promotes neuronal survival. It seems confusing regarding how Akt-mTor activity regulates neuronal survival. A better explanation would be very helpful.

4. In Figure 9, overexpression of active CRMP2 mutant was able to fully restore Pten knockout-induced optic nerve regeneration impaired by GSK3 β activation. Because GSK3 β activation blocked mTor activation, it seemed that mutant CRMP2 promoted optic nerve regeneration with impaired mTor activation. Will application of Rapamycin under the same condition still be able to inhibit optic nerve

regeneration? Moreover, if CRMP2 activation could fully rescue axon growth impaired by GSK3-SA mutants, is CRMP2 the only downstream target of GSK3 to regulate axon regeneration?

Reviewer #1: I only have few comments on how to improve the work. It would be of interest to see if the increase in GSK3 phosphorylation in control animals following ONC dependent on Rapamycin?

Response: We followed this suggestion and performed these additional experiments. Respective data are implemented as suppl. Fig. 1. A-D. The mTOR inhibition (rapamycin treatment) did expectedly not affect the ONC-induced phosphorylation of GSK3 α or GSK3 β .

Reviewer #1: A further readout for PTEN-loss is the increase in neuronal soma size. The data indicate that there might be a dichotomy on this PI3K/PTEN dependent function, which would be of great interest to the community. The authors should analyze cell size of RGC in the presence/absence of PTEN with or without the constitutive activity of GSK3 α /beta (see Figure 4).

Response: We also followed this suggestion and data are presented as suppl. Fig. 2. The RGC size was significantly increased after PTEN^{-/-} (A, B), and reduced again by constitutively active GSK3(α/β)^{S/A}. In contrast, the single knockin (GSK3 β ^{S/A}) showed no effect, suggesting that the full activity of both GSK3 isoforms is required to abrogate the PTEN^{-/-} effect. We also investigated the effect of GSK3 knockouts on the soma sizes with and without ONC (C, D). Consistently, the average soma sizes increased only after a total GSK3 knockout, whereas every single knockout alone showed no effect (C, D).

Reviewer #1: The discussion is stating the obvious that is already introduced/mentioned in the introduction/result sections. It would be of great interest to incorporate further topics, such as the potential involvement of PTEN dependent, but PI3K independent functions. Do the authors believe that all regenerative growth that can be installed through PTEN-loss involve its lipid phosphatase (i.e. is all PTEN-loss induced growth dependent on mTOR and GSK3)? Or is it conceivable that PTEN dependent, but PI3K independent function could

regulate axon regeneration.... Or that PTEN feedback regulation by GSK3 (PTEN T366) is involved in axon regeneration?

Response: We followed this suggestion and added the following sentences into the discussion part: “...without excluding any contribution of other possible, PI3K/AKT independent mechanisms, such as the influence on the activity of focal adhesion kinase (FAK), RAS or extracellular signal-regulated kinase (ERK) (Godena and Ning, 2017; Zhang et al., 2018). Whether co-expression of constitutively active CRMP2 and mTOR activation, for example by constitutively active Rheb, can fully mimic the effect of PTEN needs to be addressed in the future”

Reviewer #2: Authors applied β III-tubulin as a marker for RGC in their histology study and RGC survival quantification. β III-tubulin is a general neuronal marker but not a specific RGC marker. RBPMS and/or Brn3a should be the selective markers of ganglion cells in the mammalian retina.

Response: Although β III-tubulin is a general neuronal marker, the β III-tubulin antibody stains RGCs as specific as RBPMS in retinal flat mounts (see Figure 1 below). Moreover, in contrast to RBPMS, β III-tubulin-staining allows visualization of the cell morphology, cell size (see reviewer 1) and the retinal integrity including axons, axon bundles, and dendrites. For these reasons, we and others used β III-tubulin in this study and in our previous work e.g. (Heskamp et al., 2013; Leibinger et al., 2012; Leibinger et al., 2016; Leibinger et al., 2009; Leibinger et al., 2013; Levin et al., 2014; Sengottuvel et al., 2011). Moreover, BRN3a is only a suitable marker for non-injured RGCs. It is therefore not applicable for quantification of RGC survival after optic nerve crush since the expression of this protein is downregulated after axotomy (see our data presented in Fig. 1, below).

To demonstrate the complete overlap of RBPMS and β III-tubulin staining and the downregulation of BRN3a in axotomized RGCs to this reviewer we provide data below showing retinal flat mounts from either untreated wt mice (con) or animals 21 days after optic nerve crush (ONC) (Fig. 1, below).

Thus, additional staining with the RBPMS-antibody would not bring any additional benefit to our manuscript. Instead, repeating all survival studies would mean much additional work, delay of publication and cost many additional animal lives which appear not justifiable from an ethical point of view for this purpose.

Fig. 1: Retinal wholemounts of wild-type mice either untreated (con) or 21 days after optic nerve crush (ONC). RGCs were immunohistochemically co-stained against β III-tubulin (tubulin, red), RBPMS (green) and BRN3a (blue). Arrows indicate RGCs after ONC which are β III-tubulin- and RBPMS- positive but BRN3a- negative due to ONC-induced downregulation of BRN3a expression. Scale bar: 50 μ m.

Reviewer #2: Type error in Line 610. I should be H instead.

Response: We corrected this typo.

Reviewer #3: Addressing the following comments with either new experiments or text revising would greatly strengthen the manuscript.

In Figure 2, all the results were based on western blot of the whole retina tissue, in which proteins from RGCs were only a very small percentage of the total retina proteins. Thus, the results were not convincing enough. The authors should at least mention the potential weakness.

Response: Intravitreal injection of AAV2-Cre in PTEN-floxed mice induces the PTEN knockout specifically in RGCs (as also shown by many other studies before), because it is highly neurotropic, and RGCs are the first neurons to be encountered by the virus (Fischer et al., 2004; Leibinger et al., 2017). Reviewer #3 may have overlooked that Fig. 1 B-D and Fig. 5 B-F already show the induction of pGSK3- and pS6-levels in PTEN^{-/-} vs. PTEN^{+/+} retinæ by Western blot analysis and immunohistochemistry and thereby verify that WB analysis of retinal lysates can be used for reliable measurements in RGCs. We, therefore, feel that adding more immunohistochemical data into Fig. 2 would not add any benefit to the reader and would be rather redundant. We, therefore, would prefer to leave it as it is.

Reviewer #3: In Figure 3, the results were very similar to a previously published study, in which active Akt-induced optic nerve regeneration was impaired by GSK3b-S9A or GSK3a-S21A mutants. In Figure 3A, B, the axon lengths in many conditions were shorter than 4µm, probably less than the cell diameter. It is difficult to accurately quantify axon length with such short axons.

Response: This comment appears to be based on a misunderstanding because we provide axon lengths averaged over all RGC in the culture as described in all our previous papers and now in the M&M part. This means that the total axon length was divided by the total number of RGCs per well, including the ones that have not extended any neurite. Considering that only ~10% of RGCs extend neurites in culture, a value of 4 µm/RGC with ~500 RGCs per well (of which ~50 show neurite growth)

means that neurites of regenerated RGCs had an average length of ~40 μm ranging from 20 and 200 μm . Neurites shorter than twice the cell diameter (10 μm) were not taken into account. To clarify this issue, we added the following paragraph in the M&M section:

„The average neurite length per RGC was determined by dividing the sum of neurite length per well by the total number of RGCs per well (in average ~500 RGCs). The resulting neurite length per RGC also includes the majority of RGCs which did not grow out any neurite (~90%) thereby representing neurite growth as well as neuritogenesis. Thus, obtained values are much lower than the average axon length of RGCs with extended neurites (~20- 200 μm).”

Reviewer #3: In Figure 4, It showed that activation of GSK3a/b had no negative effects on RGC survival promoted by Pten knockout after nerve crush. This result suggests that GSK3 activation have no negative effect on Pten knockout-induced Akt-mTor activation. In Figure 5B, C, Pten knockout-induced mTor activation (increased level of pS6) was reduced by GSK3 activation (SA mutants). In Figure 5I, J double knocking out GSK3a/b increased RGCs survival by activation of mTor. Taken together, it seems that GSK3 activation reduces Pten knockout-induced mTor activation but has no negative effect on neuronal survival, whereas GSK3 knockout activates mTor and promotes neuronal survival. It seems confusing regarding how Akt-mTor activity regulates neuronal survival. A better explanation would be very helpful.

Response: This is a valuable point. We think that mTOR activity must be above a certain threshold to increase neuronal survival and soma size. This is consistent with our observation that only $\text{PTEN}^{-/-}$ and $\text{GSK3}(\alpha/\beta)^{-/-}$, which both enhanced pS6 levels to a significantly stronger extent compared to every single knockout alone ($\text{GSK3}\alpha$ or $\text{GSK3}\beta$), were neuroprotective.

As shown by the current study, $\text{PTEN}^{-/-}$ activates mTOR via two routes: Indirectly via GSK3 inhibition and directly via AKT/TSC/Rheb. Under conditions of constitutive

active GSK3 (SA mutants) mTOR activity is slightly reduced in $PTEN^{-/-}$ RGCs. However, the remaining mTOR activity seems to be still sufficiently high to confer neuroprotection. Moreover, according to our newly generated data (suppl. Fig. 2), $PTEN^{-/-}$ or $GSK3(\alpha/\beta)^{-/-}$ -induced neuroprotection correlates with an increase in RGC size (suppl. Fig. 2). While every single knockout ($GSK3\alpha^{-/-}$ or $GSK3\beta^{-/-}$) was insufficient to alter soma sizes, only the neuroprotective double knockout increased it. Moreover, in contrast to the double knockin ($GSK3(\alpha/\beta)^{S/A}$), the single knockin of $GSK3\beta^{S/A}$ failed to compromise the $PTEN^{-/-}$ induced effect on the soma size. Thus, these results suggest that mTOR activity only above a certain level increases soma size and confers neuroprotection. Nevertheless, we cannot exclude the possibility that other signaling pathways downstream from AKT, that were previously described as neuroprotective such as, e.g., NFkB might also contribute to the $PTEN^{-/-}$ effect.

To address this point we included the following paragraph into the discussion:

“The observation that $GSK3\beta^{-/-}$ failed to rescue axotomized RGCs from degeneration and did not increase soma size, while both $PTEN^{-/-}$ and the $GSK3(\alpha/\beta)^{-/-}$ induced higher pS6 levels, soma sizes and were neuroprotective suggests that a mTOR activity only above a certain level affects neuroprotection and soma size. Consistently, only the double knockin $GSK3(\alpha/\beta)^{S/A}$, but not the single $GSK3\beta^{S/A}$ knockin compromised $PTEN^{-/-}$ induced increase in the soma size, which was previously described as a consequence of S6k1 activity downstream of mTOR (Yang et al., 2014). In addition, the higher pS6 levels and soma sizes observed after $PTEN^{-/-}$ compared to $GSK3(\alpha/\beta)^{-/-}$ also correlated with a stronger neuroprotection and may be due to the additional mTOR activation via the PI3K/AKT/Rheb axis.”

Reviewer #3: In Fig. 9, the overexpression of active CRMP2 mutant was able to fully restore Pten knockout-induced optic nerve regeneration impaired by GSK3b activation. Because GSK3b activation blocked mTor activation, it seemed that mutant CRMP2 promoted optic

nerve regeneration with impaired mTOR activation. Will application of Rapamycin under the same condition still be able to inhibit optic nerve regeneration?

Response: We wish to point out that the role of mTOR was not in the focus of our manuscript, but instead the role of CRMP2. If we understand the reviewer correctly, he/she wants us to test the contribution of the remaining mTOR activity in CRMP2^{T/A} rescued regeneration. Others have previously demonstrated that rapamycin does not reduce GSK3 β ^{-/-} mediated axon regeneration (Gou et al. 2016) and we have demonstrated here and previously in Leibinger et al. 2017 that GSK3 β ^{-/-} leads to CRMP2 and mTOR activation. We, therefore, consider it very likely/predictable that inhibition of the remaining mTOR activity will not reduce regeneration. Performing these experiments would mean that we do not only need to generate data from one new group. Instead, we would have to repeat all the other comparative in vivo groups, too. We feel that this is beyond the scope of our manuscript and not feasible within three months. However, consider this point made by the reviewer we added the following sentence into the discussion part: *".... Whether inhibition of the remaining mTOR activity reduces the regenerative outcome in the presence of active CRMP2 was not addressed in the current study. However, Guo et al. have previously demonstrated that rapamycin does not reduce GSK3 β ^{-/-} mediated axon regeneration (Gou et al. 2016). As GSK3 β ^{-/-} leads to CRMP2 and mTOR activation (Leibinger et al. 2017), it is quite possible that the remaining mTOR activity is not required for the CRMP2^{T/A} effect on regeneration."*

Reviewer #3: Moreover, if CRMP2 activation could fully rescue axon growth impaired by GSK3-SA mutants, is CRMP2 the only downstream target of GSK3 to regulate axon regeneration?

Response: CRMP2 is not the only GSK3 target that is relevant for axon regeneration as shown by us previously (Gobrecht et al., 2016; Gobrecht et al., 2014). Nevertheless, in the CNS it seems to play a dominant role based on our data on CRMP2. In our

previous publication (Leibinger et al., 2017) we showed for example that active GSK3 induces MAP1B phosphorylation which does also promotes axon regeneration. However, CRMP2 activity is dominant over MAP1B as CRMP2 inhibition masked the positive effect of active MAP1B. Also, other substrates, such as APC and CLASP could be involved as well. We, therefore, did not exclude the partial contribution of other GSK3 substrates in the manuscript.

Literature:

Fischer, D., He, Z., and Benowitz, L.I. (2004). Counteracting the Nogo receptor enhances optic nerve regeneration if retinal ganglion cells are in an active growth state. *J Neurosci* 24, 1646-1651.

Gobrecht, P., Andreadaki, A., Diekmann, H., Heskamp, A., Leibinger, M., and Fischer, D. (2016). Promotion of Functional Nerve Regeneration by Inhibition of Microtubule Detyrosination. *J Neurosci* 36, 3890-3902.

Gobrecht, P., Leibinger, M., Andreadaki, A., and Fischer, D. (2014). Sustained GSK3 activity markedly facilitates nerve regeneration. *Nature communications* 5, 4561.

Godena, V.K., and Ning, K. (2017). Phosphatase and tensin homologue: a therapeutic target for SMA. *Signal Transduct Target Ther* 2, 17038.

Heskamp, A., Leibinger, M., Andreadaki, A., Gobrecht, P., Diekmann, H., and Fischer, D. (2013). CXCL12/SDF-1 facilitates optic nerve regeneration. *Neurobiology of disease* 55, 76-86.

Leibinger, M., Andreadaki, A., and Fischer, D. (2012). Role of mTOR in neuroprotection and axon regeneration after inflammatory stimulation. *Neurobiology of disease* 46, 314-324.

Leibinger, M., Andreadaki, A., Gobrecht, P., Levin, E., Diekmann, H., and Fischer, D. (2016). Boosting Central Nervous System Axon Regeneration by Circumventing Limitations of Natural Cytokine Signaling. *Mol Ther* 24, 1712-1725.

Leibinger, M., Andreadaki, A., Golla, R., Levin, E., Hilla, A.M., Diekmann, H., and Fischer, D. (2017). Boosting CNS axon regeneration by harnessing antagonistic effects of GSK3 activity. *Proc Natl Acad Sci U S A* 114, E5454-E5463.

Leibinger, M., Muller, A., Andreadaki, A., Hauk, T.G., Kirsch, M., and Fischer, D. (2009). Neuroprotective and axon growth-promoting effects following inflammatory stimulation on mature retinal ganglion cells in mice depend on ciliary neurotrophic factor and leukemia inhibitory factor. *J Neurosci* 29, 14334-14341.

Leibinger, M., Muller, A., Gobrecht, P., Diekmann, H., Andreadaki, A., and Fischer, D. (2013). Interleukin-6 contributes to CNS axon regeneration upon inflammatory stimulation. *Cell Death Dis* 4, e609.

Levin, E., Leibinger, M., Andreadaki, A., and Fischer, D. (2014). Neuronal expression of muscle LIM protein in postnatal retinae of rodents. *PLoS One* 9, e100756.

Sengottuvel, V., Leibinger, M., Pfreimer, M., Andreadaki, A., and Fischer, D. (2011). Taxol facilitates axon regeneration in the mature CNS. *J Neurosci* 31, 2688-2699.

Zhang, J., Yang, D., Huang, H., Sun, Y., and Hu, Y. (2018). Coordination of Necessary and Permissive Signals by PTEN Inhibition for CNS Axon Regeneration. *Front Neurosci* 12, 558.

REVIEWERS' COMMENTS:

Reviewer #1 (Remarks to the Author):

The authors have adressed all my concerns.

Reviewer #2 (Remarks to the Author):

The authors' response to my concern about the RGC marker in this manuscript is acceptable. I would suggest to publish this manuscript.

Reviewer #3 (Remarks to the Author):

The authors have addressed the comments with additional explanations and discussions. The study expanded our knowledge of how Pten and GSK3 signaling regulate optic nerve regeneration and the important role of CRMP2. It is now ready to be published.